# Nucleotide signaling pathway convergence in a cAMP-sensing bacterial c-di-GMP phosphodiesterase

Ian T Cadby[1], Sarah M Basford[2], Ruth Nottingham[2], Richard Meek[1], Rebecca Lowry[2], Carey Lambert[2] (iD), Matthew Tridgett[1], Rob Till[2], Rashidah Ahmad[2], Rowena Fung[2], Laura Hobley[2] (iD), William S Hughes[1], Patrick J Moynihan[1] (iD), R Elizabeth Sockett[2] (iD) & Andrew L Lovering[1,*] (iD)

## Abstract

Bacterial usage of the cyclic dinucleotide c-di-GMP is widespread, governing the transition between motile/sessile and unicellular/multicellular behaviors. There is limited information on c-di-GMP metabolism, particularly on regulatory mechanisms governing control of EAL c-di-GMP phosphodiesterases. Herein, we provide high-resolution structures for an EAL enzyme Bd1971, from the predatory bacterium *Bdellovibrio bacteriovorus*, which is controlled by a second signaling nucleotide, cAMP. The full-length cAMP-bound form reveals the sensory N-terminus to be a domain-swapped variant of the cNMP/CRP family, which in the cAMP-activated state holds the C-terminal EAL enzyme in a phosphodiesterase-active conformation. Using a truncation mutant, we trap both a half-occupied and inactive apo-form of the protein, demonstrating a series of conformational changes that alter juxtaposition of the sensory domains. We show that Bd1971 interacts with several GGDEF proteins (c-di-GMP producers), but mutants of Bd1971 do not share the discrete phenotypes of GGDEF mutants, instead having an elevated level of c-di-GMP, suggesting that the role of Bd1971 is to moderate these levels, allowing "action potentials" to be generated by each GGDEF protein to effect their specific functions.

**Keywords** bacterial signaling; *Bdellovibrio*; cAMP; Cyclic-di-GMP; EAL domain
**Subject Categories** Microbiology, Virology & Host Pathogen Interaction; Structural Biology
**The EMBO Journal (2019) 38: e100772**

## Introduction

The bacterial second messenger c-di-GMP is pivotal in controlling a variety of important cellular behaviors including virulence, motility, biofilm formation, and cell cycle progression. The machinery for c-di-GMP signaling is broadly distributed throughout bacteria, consisting in part of GGDEF synthases, EAL and HD-GYP hydrolases, named after active site motifs, and a string of sensory receptors (the PilZ domain, degenerate GGDEF/EAL proteins and specific novel proteins and riboswitches; Jenal *et al*, 2017). These signaling proteins are usually multidomain in nature, coupling a sensory input to the requisite c-di-GMP utilizing domain(s). Structures exist for EAL proteins from several organisms, most notably active forms of the blue-light photoreceptor BlrP1 from *Klebsiella pneumoniae* wherein allosteric control appears to be afforded through domain:domain interfaces (Barends *et al*, 2009). Activation of EAL proteins via dimerization around a helix-rich interface appears to be a widespread mechanism for regulation, altering the conformation of metal-coordinating residues at the active site (Dahlstrom & O'Toole, 2017). Nevertheless, no high-resolution structures exist for c-di-GMP hydrolases in both the active and inactive states. Here we investigate the allosteric sensory regulation of the single EAL domain protein present in the bacterial predator *Bdellovibrio bacteriovorus*.

*Bdellovibrio* (and like organisms) are predators of a range of other bacteria and include *B. bacteriovorus* (in the δ-proteobacteria grouping), which has evolved the remarkable adaptation of killing its prey via periplasmic invasion (Sockett, 2009). Sequentially, predatory life cycle stages involve prey detection, adhesion, recognition, breach of outer membrane, periplasmic invasion, membrane resealing, hydrolysis and assimilation of prey macromolecules, growth of a single filamentous predator cell, septation into progeny, and final lysis of prey cell and escape. Predatory bacteria have been proposed and trialed in model systems for use in antibacterial applications in both health care and agriculture, and we are beginning to understand the novel processes behind regulation of predation as a lifestyle (Negus *et al*, 2017). *Bdellovibrio bacteriovorus* were described as having a high c-di-GMP "intelligence" (Galperin *et al*, 2010), encoding at least 18 PilZ proteins and tens of putative novel receptors (Rotem *et al*, 2015). The high density of c-di-GMP targets is in contrast to the lower number of four active and one degenerate GGDEF synthases identified.

1   Institute for Microbiology and Infection, School of Biosciences, University of Birmingham, Birmingham, UK
2   Centre for Genetics and Genomics, School of Biology, Medical School, Queen's Medical Centre, Nottingham University, Nottingham, UK
    *Corresponding author. Tel: +44 121 41 45419; E-mail: a.lovering@bham.ac.uk

Disruption of c-di-GMP production via individual GGDEF gene knockouts results in discrete *B. bacteriovorus* phenotypes (obligate predation/obligate axenic growth/slower invasion of host/compromised gliding motility; Hobley *et al*, 2012). The lack of compensatory overlap between GGDEFs (in a small bacterium where diffusion would be expected) is demonstrative of precise signaling, as opposed to broad activation from a common freely available pool of c-di-GMP. Candidates for lowering of c-di-GMP levels in *Bdellovibrio* are limited: Only two of the six noted HD-GYP proteins have the consensus motif for substrate binding (Lovering *et al*, 2011), and homology searches identify a single EAL domain phosphodiesterase, Bd1971. HD-GYP enzymes are capable of converting c-di-GMP into GMP, whereas EAL enzymes linearize it into pGpG which may act as an additional signaling agent before it is hydrolyzed via small oligoribonucleases (Jenal *et al*, 2017). The differential behaviors of these two types of phosphodiesterase may provide an additional level of complexity in that (in *P. aeruginosa*) pGpG extends the half-life of circulating c-di-GMP via EAL inhibition (Orr *et al*, 2015). The Bd1971 domain architecture is predicted to consist of an N-terminal cNMP sensory domain (cyclic nucleotide monophosphate, postulated to bind cAMP or cGMP), a middle linker region, and C-terminal catalytic EAL barrel.

Herein, we characterize the relationship between the sensory cNMP domain and catalytic EAL domain of Bd1971, determining high-resolution structures of the protein in multiple states. We reveal that regulation of c-di-GMP hydrolysis arises from a domain-swapping of the "regular" cNMP fold, which undergoes significant rearrangements upon binding of stimulus, communicating a series of conformational changes over 40 Å to license a productive state at the EAL dimer interface.

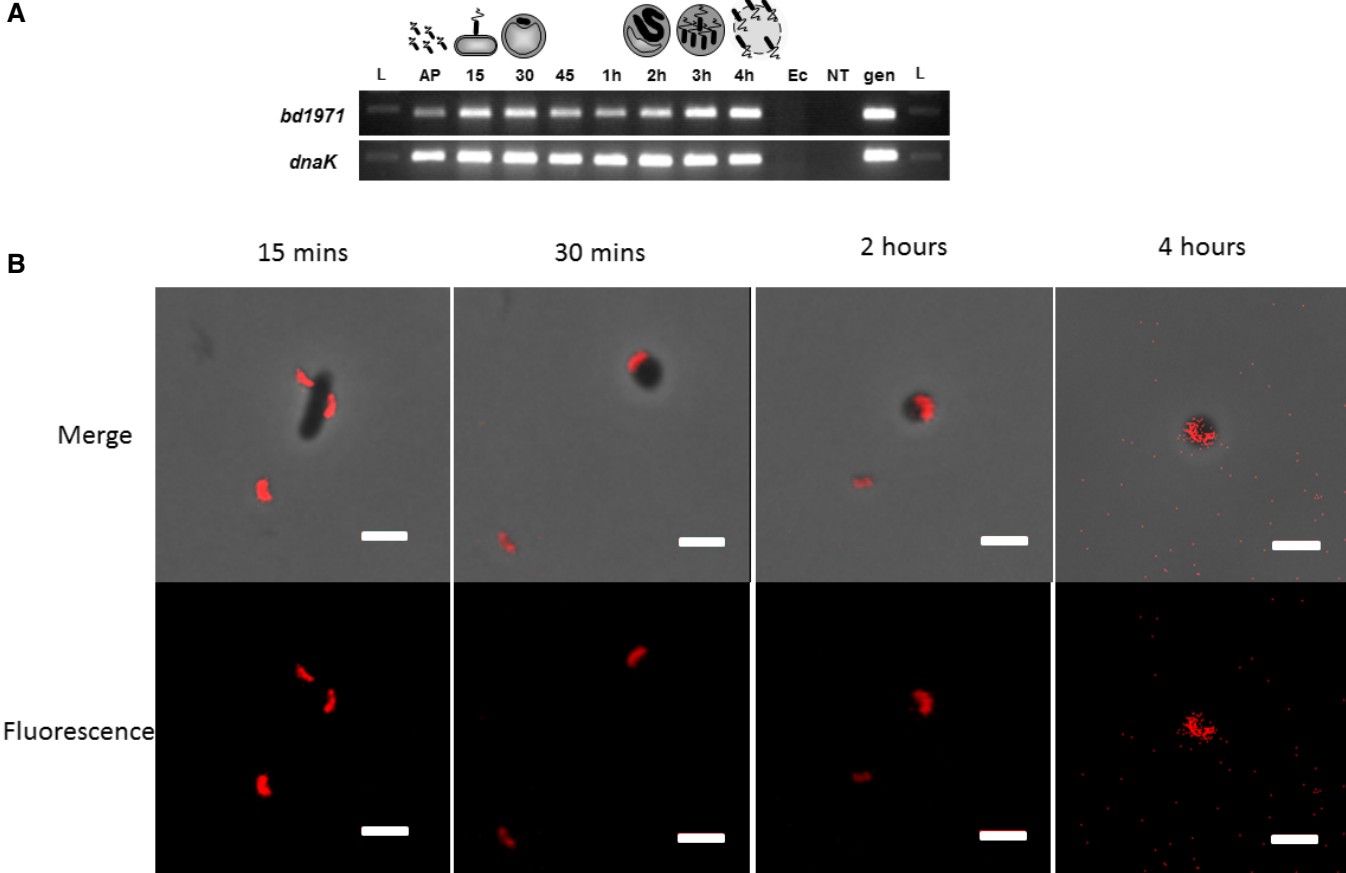

**Figure 1. Bd1971 Expression and Localization During the *Bdellovibrio* Life cycle.**

A   Reverse transcriptase PCR showing that *bd1971*, and control gene *dnaK*, are expressed throughout the predatory cycle of *Bdellovibrio bacteriovorus*. RNA was isolated at the timepoints indicated across the top of the gel during one round of synchronous *Bdellovibrio* infection of *E. coli* cells. Primers were designed to anneal specifically to the gene of interest. L = 100 bp DNA ladder, AP = Attack Phase cells, 15–45 = 15–45 min respectively since infection, 1–4 h = hours respectively since infection. Ec = *E. coli* strain S17-1 RNA (negative control: no *Bdellovibrio* RNA); NT = control with no template RNA; gen = *B. bacteriovorus* HD100 genomic DNA (positive control). The cartoon above represents each stage in the predatory cycle.

B   Epifluorescent images of *E. coli* invaded by *B. bacteriovorus* HD100 with mCherry tagged Bd1971 at the times indicated throughout the invasive process during one round of synchronous infection. Fluorescence was acquired with a two-second exposure and maximum sensitivity gain, but different balance and contrast levels in the fluorescence channel were used to produce these images (see Materials and Methods) as the fluorescence was fainter at later timepoints but we wished to show whether or not the cellular localization within the *Bdellovibrio* altered during predation. Fluorescence images and merges with phase contrast images are shown. Scale bars are 2 μm. Images are representative of three independent experiments.

# Results

## Bd1971 is expressed throughout the predatory life cycle and shows diffuse localization

As the only putative EAL domain containing protein in *Bdellovibrio*, Bd1971 presents a potential means of terminating c-di-GMP signaling. Building on previous work demonstrating the importance of GGDEF proteins in regulating the *Bdellovibrio* predatory life cycle, we sought to determine the role of Bd1971. RT–PCR on RNA extracted throughout the predatory life cycle showed that *bd1971* is expressed at all timepoints (Fig 1A). In agreement with this, fluorescent tagging of Bd1971 with a C-terminal tag of mCherry revealed fluorescence at all points throughout the predatory cycle, albeit slightly fainter at later timepoints (Fig 1B). This fluorescence was diffuse throughout the predator cell rather than localizing to specific subcellular foci. These results indicate that *bd1971* is expressed constitutively throughout the *Bdellovibrio* life cycle and, since Bd1971 lacks any recognizable signal peptide, the protein is located throughout the cytoplasm.

## Disruption or deletion of *bd1971* results in higher global c-di-GMP levels

In order to determine the biological function of Bd1971, we constructed mutant strains of *Bdellovibrio*. Initially, we constructed a kanamycin insertion mutant of *bd1971* and observed this strain to have higher global levels of c-di-GMP (Fig 2A). We then constructed a silent deletion strain, a strain Bd1971D306307A with both catalytic aspartate residues exchanged for alanine and a strain Bd1971R67D, with the arginine of the cNMP-binding pocket exchanged for aspartic acid. These strains also had higher global levels of c-di-GMP than the wild type (Fig 2B), suggesting that this was a result of the loss of phosphodiesterase activity.

## Bd1971 structure reveals juxtaposition of sensory and catalytic domains

Having observed perturbed c-di-GMP levels for the *bd1971* mutants, we sought to structurally and biochemically characterize the Bd1971 protein and confirm its role as bona fide c-di-GMP phosphodiesterase. The crystal structure of full-length Bd1971 was solved with both cAMP and cGMP ligands, to resolutions of 2.63 and 2.46 Å, respectively (Table 1). These two co-crystal structures are effectively identical other than small differences at the ligand binding pocket that are discussed later. Bd1971 forms a butterfly shaped dimer measuring 90 Å tall, 90 Å wide (at larger EAL domain) and 40 Å deep (Fig 3A–C). The central twofold axis acts to situate an EAL domain dimer (residues 158–402) over a cNMP sensor dimer (6–111). The two globular folds of these subdomains are separated by an extension to the cNMP β-sandwich (residues 117–123, linear in conformation) and a long linker helix (136-157). The linear extension is only resolved in the higher resolution cGMP dataset and has relatively weak electron density for side chain residues; we have estimated the amino acid register in this region from physical attributes and conservation patterns of the segment, but it remains ambiguous. Omit maps for the linker and several ligands are provided in Fig EV1. The sections of the polypeptide chain we are unable to observe (flanking the β-extension) are

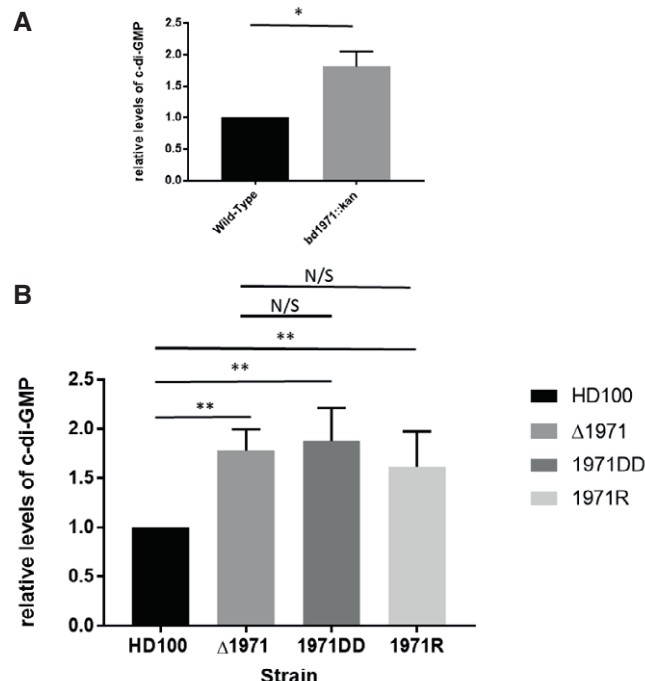

**Figure 2. c-di-GMP Levels in Bdellovibrio and bd1971-variant Strains.**

A Relative levels of c-di-GMP in matched cell biomass host-independent (HI) cultures of *bd1971::kan* compared with *fliC1* merodiploid Kn[R] HI control strain (wild type for *bd1971*). The *bd1971* mutant strain contains significantly more c-di-GMP than the control. Data are from 3 independent experiments.

B Relative levels of c-di-GMP in matched cell biomass wild-type predatory cultures of HD100 strain, silent Bd1971 deletion strain (Δ1971), Bd1971D306307A strain (bd1971DD), and Bd1971R67D strain (bd1971R). All mutant strains had higher levels of c-di-GMP compared to the wild-type strain. Error bars show SEM. Data are from six independent experiments for HD100 and ΔBd1971 and from four independent experiments for 1971DD and 1971R mutants.

Data information: Error bars show SEM. *$P < 0.05$ **$P < 0.01$ by Student's *t*-test. **$P < 0.01$ by Mann–Whitney *U*-test.

Source data are available online for this figure.

glycine-rich, with little conservation in homologues. The cNMP sensor region can be subdivided into the main body of the fold (6–83), a hinge helix (84–95) and a central, dimer-forming helix (96–111, named the C-helix in relation to cyclic AMP receptor protein, CRP; Passner *et al*, 2000). Restrictions provided by both crystal packing and the relative positions of the C-helix end, extension and linker helix start are suggestive of a domain swap (the distance between residues 111 and 117 is a more favorable 11 Å in a domain swap and a less favorable 21 Å for non-swapped assignment) such that chains A and B have the cNMP sensor and EAL catalytic domains on opposing sides of the dimer. The juxtaposition of sensor and catalytic domains (cNMP domain closer to EAL barrel start, at the open face of active site) is akin to that of the GGDEF domain of MorA (Phippen *et al*, 2014), and opposite to that of the closed-face/dimer interface location of the BlrP1 BLUF sensor (Barends *et al*, 2009) or RocR trans-REC sensor (Chen *et al*, 2012).

There are no other structures of a cNMP:EAL hybrid protein, but separate structure comparisons using DALI confirm homologues for the individual subdomains of Bd1971—the EAL αβ(βα)7

**Table 1. Data collection and refinement statistics.**

| | Full-length + cGMP | Full-length + cAMP | Full-length + cAMP + c-di-GMP | 1-150 Bd1971$^{\Delta EAL}$ Apo-form | 1-150 Half-site + cAMP |
|---|---|---|---|---|---|
| Accession code | 6HQ7 | 6HQ4 | 6HQ5 | 6HQ2 | 6HQ3 |
| **Data collection** | | | | | |
| Space group | P3$_1$ | P3$_1$ | P3$_1$ | P4$_1$2$_1$2 | C2 |
| Cell dimensions | 82.6, 82.6, 134.6 | 82.4, 82.4, 131.3 | 83.9, 83.9, 138.5 | 92.5, 92.5, 74.1 | 94.6, 120.8, 56.3 |
| $a, b, c$ (Å) | | | | | |
| α, β, γ (°) | 90, 90, 120 | 90, 90, 120 | 90, 90, 120 | 90, 90, 90 | 90, 116.8, 90 |
| Resolution (Å) | 2.46 (2.52-2.46)$^a$ | 2.63 (2.7-2.63) | 2.83 (3.06-2.83) | 2.45 (2.51-2.45) | 2.79 (2.86-2.79) |
| $R_{merge}$ | 3.8 (52.7) | 4.1 (78.9) | 5.7 (46.2) | 5.2 (-) | 6.7 (74.9) |
| $R_{pim}$ | 2.9 (38.9) | 3.6 (69.3) | 5.1 (41.6) | 2.2 (75.0) | 6.4 (70.5) |
| CC 1/2$^b$ | 99.9 (78.3) | 99.9 (61.6) | 99.8 (77.7) | 1.00 (81.5) | 99.6 (51.8) |
| $I/\sigma I$ | 18.9 (2.9) | 14.3 (1.4) | 9.3 (1.9) | 23.9 (1.6) | 8.3 (1.1) |
| Completeness (%) | 99.9 (100.0) | 99.5 (99.8) | 99.7 (99.6) | 99.5 (99.4) | 98.5 (99.3) |
| Redundancy | 5.2 (5.4) | 4.0 (4.2) | 4.0 (4.0) | 12.5 (12.0) | 3.2 (2.7) |
| **Refinement** | | | | | |
| Resolution (Å) | 2.46 | 2.63 | 2.83 | 2.45 | 2.79 |
| $R_{work}/R_{free}$ | 21.9/25.1 | 24.0/28.6 | 22.3/26.8 | 20.9/24.9 | 21.4/25.9 |
| R.m.s. deviations | | | | | |
| Bond lengths (Å) | 0.008 | 0.006 | 0.006 | 0.014 | 0.010 |
| Bond angles (°) | 1.32 | 1.08 | 1.07 | 1.76 | 1.59 |

$^a$Values in parentheses are for highest-resolution shell.
$^b$CC 1/2 is the correlation coefficient between two random half data sets.

barrel matches the well-characterized EAL from *T. denitrificans* (Tchigvintsev *et al*, 2010; 3N3T, 1.7 Å RMSD for 254AA alignment, 35% sequence identity). The cNMP sensor region matches equivalent regions in both CRP-like proteins (*M. tuberculosis* 3MZH, 3.5 Å RMSD, 105AA, 29% identity) and protein kinases (*P. falciparum* 5KBF, 2.0 Å RMSD, 97AA, 31% identity). Currently, EAL dimers can be subclassified into three states based on the nature of the interacting faces—"open", "closed", and "offset" (Bellini *et al*, 2017). Bd1971 dimerizes around a long helix (residues 340–357, Fig 3B) and a shorter helix (315–321, referred to as the compound helix in BlrP1 or repressor helix in MorA; Phippen *et al*, 2014) that follows the catalytic aspartate pair (Asp306/307), and can thus be assigned to the "open" grouping that correlates with active enzymes. The Bd1971 dimer interface is non-crystallographic, and so does not display perfect twofold symmetry—region 311–323 in chain B is more mobile than chain A, and we have modeled this in two (related) conformations. The dimerization interface we observe is centered around Ser348 and places Asp306 and Asp307 in a productive orientation, similar to that observed in other (presumed active) EAL dimers (Fig 3D).

## Bd1971 in cNMP stimulus-occupied state forms a consensus active dimer, pre-configured to bind c-di-GMP substrate

Our observation of a classical open/productive EAL dimer for Bd1971 led us to investigate competency for c-di-GMP binding at the active site. Crystals of Bd1971 (grown using cAMP ligand) were soaked in solutions supplemented with c-di-GMP and Ca$^{2+}$

(included to prevent substrate turnover). Clear electron density was observed for c-di-GMP at both dimer active sites, bound in the expected extended conformation (determined to 2.83 Å resolution, structure in Fig 3E). In comparison with our multiple structures with an empty EAL active site, there are no gross changes upon binding substrate; this is distinct from MorA, whose H-helix (mapping to the Bd1971 interdomain linker helix) and associated GGDEF domain undergo a 20° rotation when c-di-GMP binds (Phippen *et al*, 2014). Smaller, local changes upon binding are limited to the metal-binding pocket (which like other EAL domains, binds one metal in the empty state and two metals in the substrate complex) (Dahlstrom & O'Toole, 2017), and residues which undergo rotameric changes to accommodate c-di-GMP (Arg330 shifts to relieve a clash with the nucleotide base and ribose, Arg182 and Tyr385 move inward to interact with base and phosphate, respectively). The configuration of the active site residues of Bd1971 is similar to other (active) EAL structures, further indicating that this form represents the active state of the Bd1971 EAL domain; hence, we instigated further study into how the novel domain structure (with a stimulus-sensing cNMP domain) could switch EAL activation states.

## The sensor domain of Bd1971 is an adapted variant of the classical CRP-like cNMP dimer with unique domain-swapped symmetry

The cNMP sensory module, at the N-terminus of Bd1971, is used in a variety of contexts in other proteins, the better-characterized examples being transcriptional regulators (exemplified by CRP from

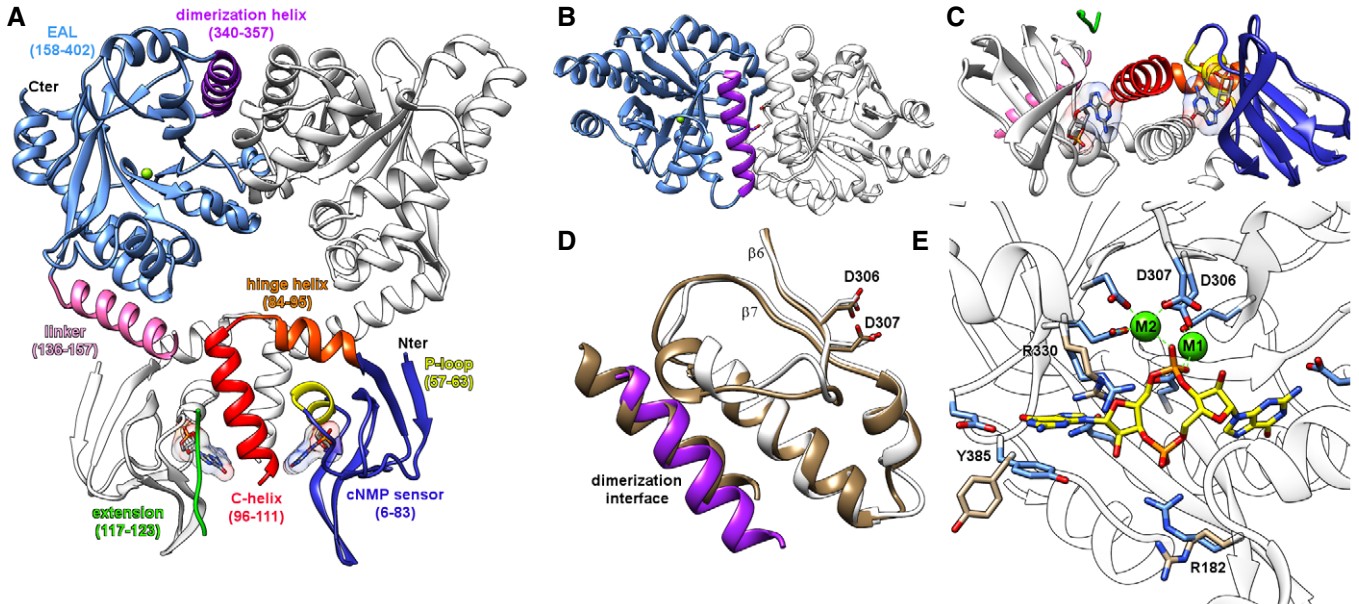

**Figure 3. Fold of the composite cNMP-EAL Bd1971.**

A   Dimer of full-length cGMP-bound Bd1971; one monomer colored individually by subdomain, the opposing monomer colored white. Ligand represented in stick form, EAL active site Mg$^{2+}$ as green sphere.

B   View from top of that in (A), detailing EAL region of dimer interface.

C   View from opposing side, detailing C-helix dimer interface at cNMP domain.

D   The Bd1971 EAL dimerization interface (purple, white) matches that of a canonical, active dimer from the well-characterized *T. denitrificans* EAL (tan, PDB 2r6o); the catalytic Asp pairs of the two enzymes adopt a similar conformation (stick form, D306/307 from Bd1971, D646/647 from 2r6o).

E   Structure of the cyclic-di-GMP bound complex of Bd1971 (ligand C represented in yellow, bound Ca$^{2+}$ ions as green spheres). Residues R182, R330, and Y385 display the largest shift between the apoenzyme (tan) and bound (blue) states, R182 complexing the cyclic-di-GMP phosphate, and Y385 stacking with a guanine base.

*E. coli*) and cNMP-dependent protein kinases. Structural comparisons indicate that, although the Bd1971 N-terminus is more homologous to the cNMP domains of selected protein kinases, the cNMP:cNMP contacts of the Bd1971 dimer more closely resemble the functional arrangement of CRP (Fig 4A–C). CRP proteins use dimerization around a central C-helix to position and regulate a DNA-binding domain via C-helix extension (Passner *et al*, 2000). The location of the CRP DNA-binding domain is antipodal to the Bd1971 EAL domain and suggestive of an entirely different mode of regulation/activation (Fig 4A). Despite this difference, the sensory domains of the Bd1971 cNMP dimer and CRP cNMP dimer superimpose well, with an RMSD of 1.8 Å (common atoms, 180 residues) for the cAMP-bound *E. coli* archetype.

The clear agreement of the Bd1971 and CRP dimers is notable given that the C-helices of the twofold are in spatial agreement but originate from different monomers; the classic CRP arrangement places the C-helix closest to the β-sandwich of the same sensor, whereas in Bd1971 the C-helix is "swapped over", lying closest to the sensor pocket of the opposing monomer (Fig 4B). To the best of our knowledge, this arrangement is unique among all determined cNMP-containing protein structures, and results from a different packing angle of the hinge helix relative to the dimer interface (Fig 4B).

Superimposition using the cAMP ligand reveals many shared residues/structural features between Bd1971 and CRP (Fig 4C) including the classical base capping residue (Bd1971 Arg108: CRP Arg123), phosphate cradle (Arg67/Ser68: Arg82/Ser83), and

ribose-coordinating group (Glu58:Glu72). Due to the relative C-helix swapping, the hydrophobic base capping residues (Arg108 and Leu109) are contributed from the opposing monomer in Bd1971 and the same monomer in CRP (Fig 4C). A secondary Bd1971-specific hydrophobic interaction with the adenine is provided by Met59, which sits against both the base and the ribose of cAMP.

All functional cNMP sensory domains contain a region between two β-strands that forms a small α-helix and loop region (called the P-loop or phosphate binding cassette, PBC) that responds to the nucleotide sugar phosphate. In Bd1971, the P-loop sequence (usually given as a longer 14AA window, but shortened to the helix alone in Fig 3A) matches the superfamily consensus well, with the sequence F[56]GEMALIDNQNRSA[69]. The residues responsible for recognizing the edge of the cAMP adenine ring in *E. coli* CRP (Thr127 and Ser128) are replaced by Asn113 in Bd1971; other CRP proteins (e.g., 4CYD from *C. glutamicum*) can also use Asn for this interaction (Townsend *et al*, 2014), although we note that wider Bd1971 homologues display some sequence variation at this residue.

### The phosphodiesterase activity of Bd1971

Next, we sought to determine experimentally whether Bd1971 has phosphodiesterase activity. Since the ability of the HTH domains of CRP to interface with DNA is regulated by its cNMP domains, we postulated that binding of cAMP or cGMP to the cNMP domains of Bd1971 could have a similar allosteric effect on phosphodiesterase activity. Hydrolysis of the general phosphodiesterase substrate

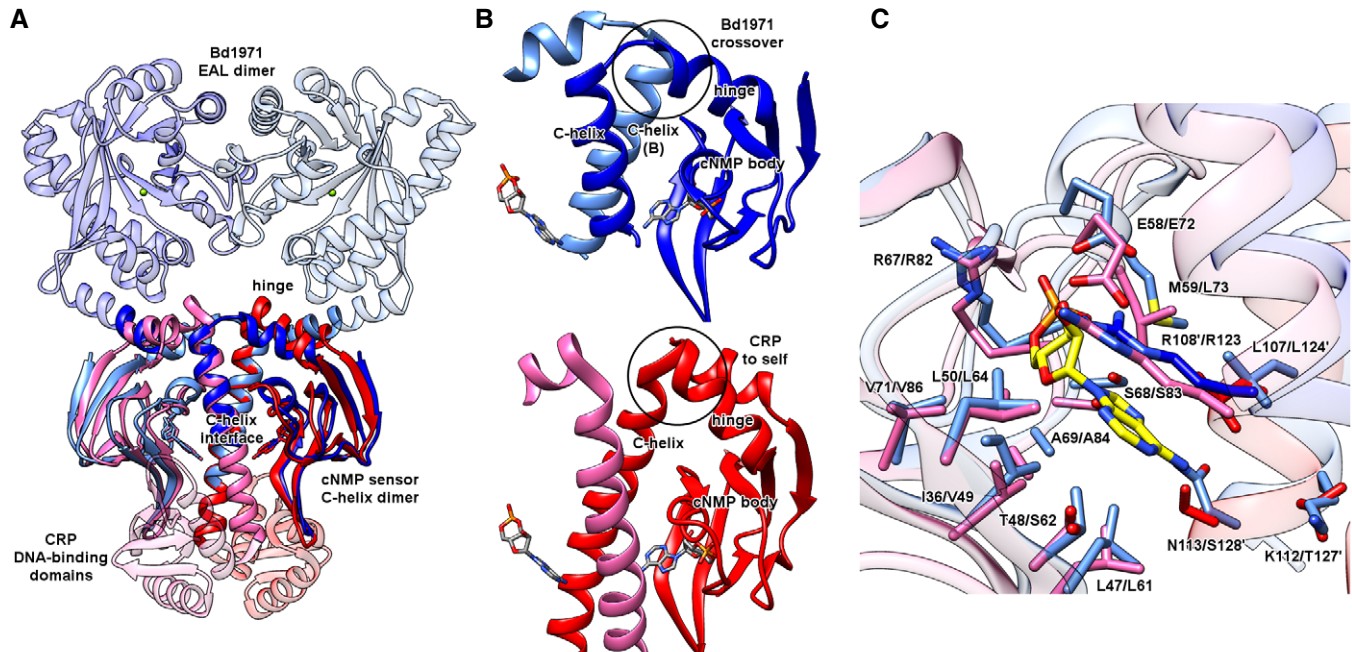

**Figure 4. Comparison of Bd1971 cNMP dimer to that of CRP (from *E. coli*, PDB 1g6n).**

A  The Bd1971 cNMP dimer (light/dark blue) situates its output domain (EAL) on the opposite side to that of CRP (DNA-binding subdomain, dimer colored pink/red). Both Bd1971 and CRP dimerize via a central C-helix:C-helix interface.

B  Close-up of hinge helix to C-helix crossover (circled). Tilt of the hinge helix relative to the C-helix allows Bd1971 to bridge from the cNMP domain body into the C-helix of the opposing monomer; the arrangement in CRP differs by pairing the C-helix with the cNMP body of the same monomer. Despite this difference, the respective C-helix dimers remain in the same orientation, as observed in panel (A).

C  Overlay of the cAMP-binding pocket of Bd1971 and CRP (aligned on cAMP ligand, Bd1971 residue numbering given first), ligand-contacting side chains shown in stick form, residues from the opposing monomer labeled by a'.

---

$p$-nitrophenyl phosphate (PNPP) by purified apo-Bd1971 supplemented with either $Mg^{2+}$ or $Ca^{2+}$ and an excess of cAMP or cGMP was assayed (Fig 5A).

Control apo-Bd1971 in the absence of cNMP had negligible activity toward PNPP, as did apo-Bd1971 supplemented with cNMP but lacking $Mg^{2+}$. The presence of $Ca^{2+}$ also failed to stimulate activity. Phosphatase activity, shown as PNPP conversion, by Bd1971 was detected in the presence of $Mg^{2+}$ and either cAMP or cGMP, indicating that cNMPs function as activators of $Mg^{2+}$-dependent Bd1971 phosphodiesterase activity. The maximal phosphodiesterase activity stimulated by cGMP was 22% higher than that of cAMP.

We next tested whether Bd1971 has activity toward the assumed native substrate, c-di-GMP. Reaction mixtures containing c-di-GMP and either apo- or cNMP-supplemented Bd1971 were resolved by HPLC (Fig 5B). Consistent with our predictions, Bd1971 converted c-di-GMP to pGpG. C-di-GMP turnover was increased ~8-fold in the presence of cAMP or cGMP, further supporting the proposal that cNMPs are activators of Bd1971.

### Structure of a truncation mutant reveals large-scale changes associated with an empty sensor domain

We next sought to rationalize how cNMPs regulate Bd1971 phosphodiesterase activity. The large distance between the sensor cAMP pockets and EAL active sites (40 Å between chains, 50 Å within the same chain) required us to understand the means by which long-range allostery could gate Bd1971 activity. Despite exhaustive attempts, we were unable to stabilize full-length apo-Bd1971 at concentrations high enough for biophysical characterization (concentrated preparations of full-length apo-Bd1971 protein formed heavy precipitates in the longer time-scale of crystallization trials and isothermal calorimetry experiments although it should be noted that these precipitates would partially re-dissolve on addition of cAMP or cGMP). To gain insight into the mode of allosteric regulation, we therefore designed a truncation mutant (residues 1–150, referred to hereafter as Bd1971$^{\Delta EAL}$, truncated at the point which the linker helix no longer contacts the cNMP sensor domain) that was amenable to study. The structure of the Bd1971$^{\Delta EAL}$ construct was determined to 2.45 Å resolution, enabling us to trace residues 4–111 of the sensor-hinge–C-helix region of the protein (Fig 6). This truncated variant retains the CRP variant dimer interface observed in the full-length protein and has two clearly empty cNMP sensor domains.

The Bd1971$^{\Delta EAL}$ structure reveals that the relationship between the cNMP body and hinge/C-helix dimer interface has changed relative to that of the full-length cAMP-bound form, such that the apo/bound structures can be superimposed on one of these elements but not both. This conformational change upon losing ligand is best described by the realization that the C-helix interface is essentially unchanged (between apo and bound states), and acts as a stator to allow the cNMP body to swing-out relative to the central twofold

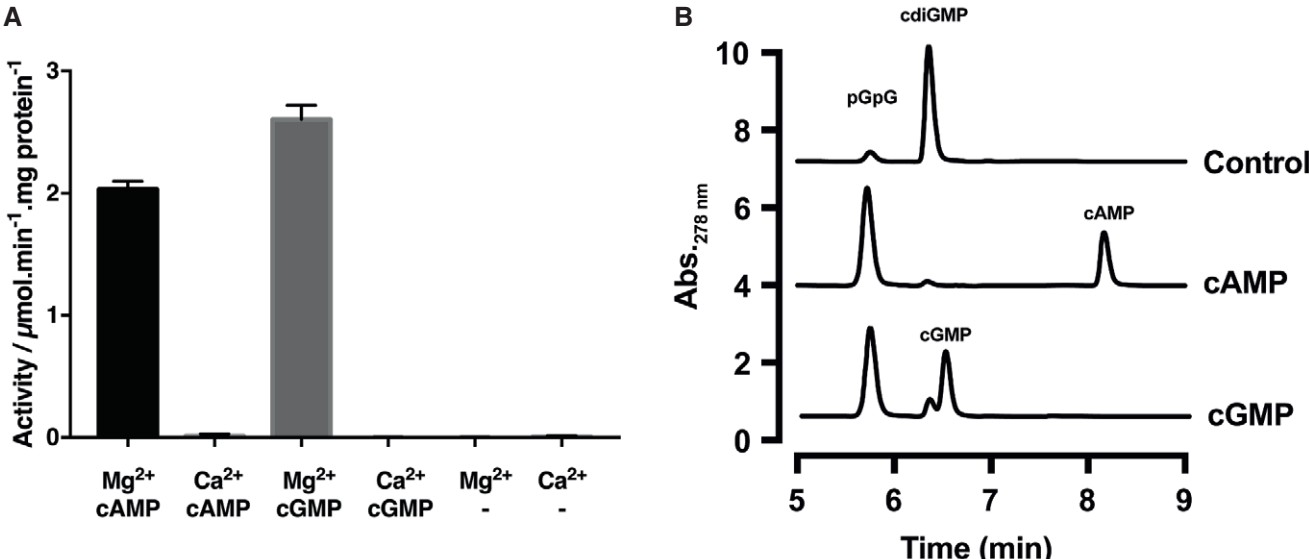

**Figure 5. Phosphodiesterase activity of Bd1971 is stimulated by the allosteric effectors cAMP and cGMP.**

A   Apo-Bd1971 was supplemented with 10 mM MgCl$_2$ or CaCl$_2$ in conjunction with saturating concentrations (2.5 mM) of cAMP, cGMP, or no nucleotide and phosphodiesterase activity with respect to the generic substrate PNPP (5 mM) was measured spectrophotometrically. Values are the average of three repeats, and error bars represent standard deviation.

B   HPLC traces of reaction products generated from 100 μM c-di-GMP incubated with apo-Bd1971 (Control), or protein supplemented with either 100 μM cAMP or cGMP. Peaks were identified by reference to standards also resolved by HPLC and are labeled in the figure.

Source data are available online for this figure.

axis (Fig 6A–D). Alignment of the bound and apo-forms using the C-helix interface (residues 97–110 of both chains) reveals that the cNMP body rotates ~50° relative to the plane between the C-helices. Residues 81-86 spanning the connection between the cNMP body and hinge helix comprise the fulcrum for this motion (identified using DynDom; Girdlestone & Hayward, 2016, labeled "cNMP link" in Fig 6B and C), resulting in the largest translation between apo and bound structures occurring at the distal end of the fold (the β-hairpin element of the cNMP body shifts 25 Å, as depicted by an arrow in Fig 6A). The concerted rotation and translation of the cNMP fold places the outer face of the β-sandwich (strands 1, 3, 6, and 8) into an orientation that would sterically clash with the linker helix of the opposing chain in the cAMP-bound full-length structure (Fig 6A).

Classical cNMP transitions between apo and bound forms (in a variety of other protein folds) are known to use changes in the P-loop upon recognizing the nucleotide phosphate to drive conformational rearrangements (Kornev *et al*, 2008). The Bd1971-bound and Bd1971$^{\Delta EAL}$ apo structures display a large P-loop movement, relative to a hydrophobic pocket created by the hinge/C-helix residues Val87, Leu88, Val91, Met102, and Leu106 (Fig 6B and C). In the bound form, the P-loop is in an "up" conformation, with residues Leu61 and Ile62 forming an interface with the hydrophobic pocket; this is further stabilized by a bifurcated interaction between Arg110 and Asp63 of opposing chains (Fig 6C). In the apo-form, the P-loop migrates away from this pocket, with Asp63 sat below Arg110, breaking the aforementioned interaction (Fig 6B). Superimposition using the cNMP body rather than the C-helices informs on the nature of this relative P-loop shift (Fig 6D); the P-loop α-helix

position is constrained in the bound form by hydrogen bonding between the backbone amide of Ala60 and the cAMP phosphate. Upon the transition to an apo-form, the P-loop twists, placing Leu61 in an orientation incompatible with bound-state hinge residues Val87 and Leu88, hence driving a different juxtaposition of the cNMP body and C-helices (Fig 6D). Bd1971 is unusual in that Val87 sterically replaces a hinge residue that is usually Phe or Tyr in cNMP sensors (Kornev *et al*, 2008).

We observe that, in addition to this classic P-loop mediated repacking, a secondary factor contributes to the transition between "tight"/cNMP inward and "loose"/cNMP outward states. The central C-helix interface is formed in part by residues Leu105 and Leu106 from both chains (Fig 6E). Residue Met59 of Bd1971 provides a secondary cAMP-responsive component in that its side chain forms a stacking interaction between the nucleotide base and Leu106, allowing the cNMP body to closely approach both C-helices (Fig 6F). In the apo-form, Met59 is in a different rotameric conformation, unable to now stack with Leu106 (moving ~8 Å away to stack with Arg108 of chain B, Fig 6B), assisting domain swing-out.

### Comparison of the cNMP bound and apo structures of Bd1971 reveals a second region of hydrophobic remodeling at the interdomain linker

Observation of steric clash between the apo-form cNMP body and bound form interdomain linker helix led us to examine the interactions that support active EAL domain orientation in the bound state. Importantly, the linker helix (residues 136–157) is central to the productive full-length bound form, contacting three

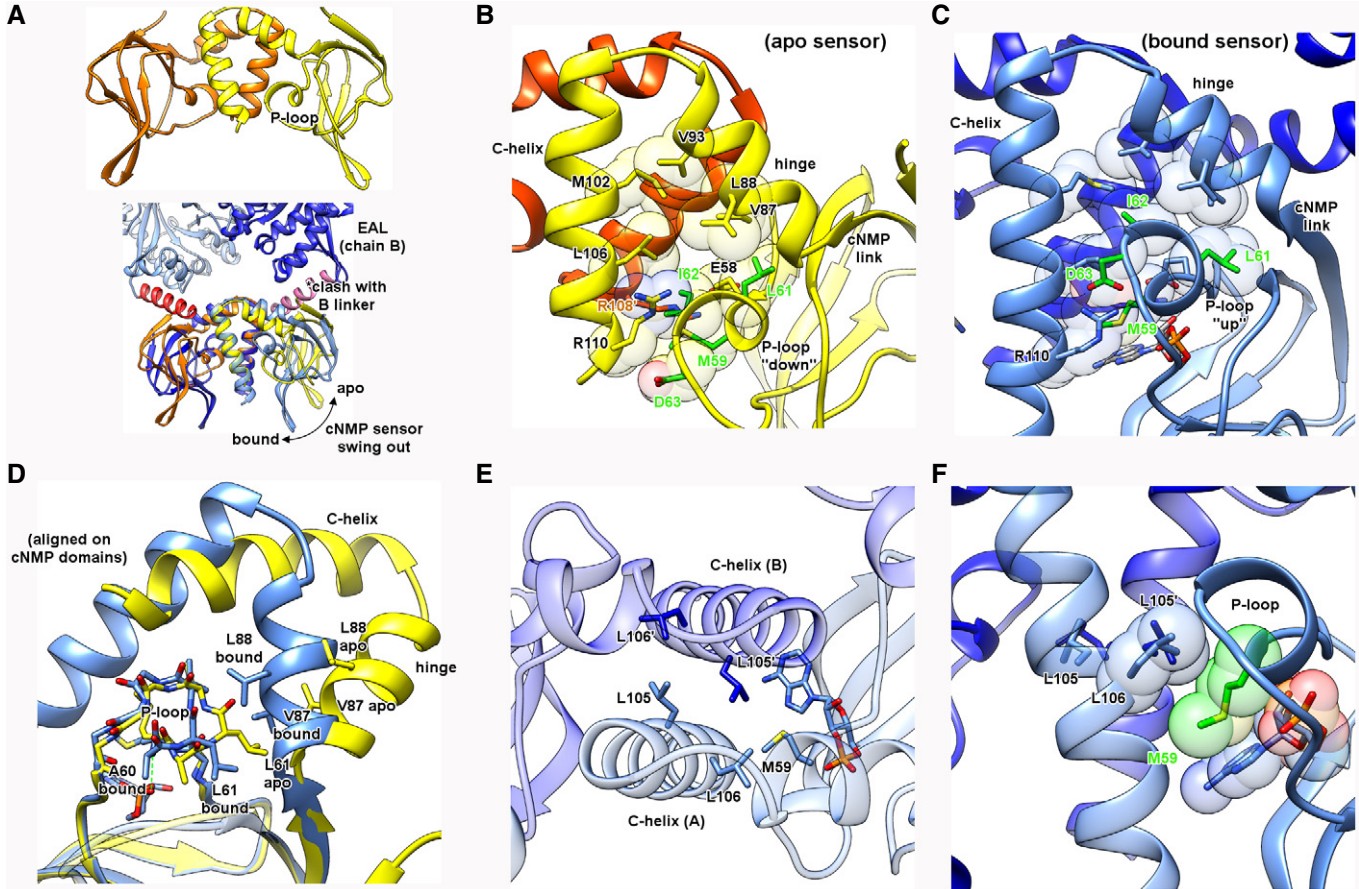

**Figure 6. Structure of an apo-form of the cNMP sensor.**

Structure derived from a Bd1971$^{\Delta EAL}$ construct (yellow/orange), in comparison with cAMP-bound full-length structure (dark/light blue, interdomain linker helices in red/pink). Residues from the opposing monomer labeled by a'.

A    The apo sensor overlaid with the cAMP-bound sensor indicates a relative swing-out of the cNMP body relative to C-helix central stator (motion indicated by arrow); the apo-form would now sterically clash with the linker helix in the full-length bound structure.

B, C   Comparison of P-loop environment in both apo and bound states. In the apo-form, the P-loop (selected residues in green) migrates out of a hydrophobic pocket formed by the hinge and C-helices, breaking an interaction between R108' and D63. In the bound state, M59 stacks with the nucleotide base and the P-loop shifts upward to contact both sides of the pocket.

D    An alignment via the cNMP body demonstrates that swing-out is licensed in part by P-loop twisting/migration—L61 in the bound state allows close packing of the hinge and C-helices, whereas L61 in the apo state would clash with V87 and L88, forcing a relative shift of the body and helical subdomains.

E, F   Engagement of the cAMP base and M59 with the C-helix interface. The two orthogonal views demonstrate that L105 and L106 form a hydrophobic plane at the C-helix interface, under which the M59 side chain can pack against the cAMP base and allow the cNMP body to adopt the swung in conformation.

separate domains—both cNMP sensors (one chain via the N-terminal end of the C-helix, the other chain via the outward face of the β-sandwich), and the EAL domain (at the αββα start of the barrel, Fig 7A). The side chains of the linker helix present a mix of both polar and hydrophobic interactions to the globular domains, with only two residues from the helix (Gln138 and Asp142 on the outer face) not making significant contact with either cNMP or EAL residues.

Through comparing the apo and cNMP-bound structures of Bd1971, the cNMP domain swing-out appears to be licensed by the hydrophobic face, comprised of linker helix residues Ala140, Leu141, and Ile144 (Fig 7B). In the bound state, these three residues serve to provide a "hydrophobic wedge" situated between Val97 and Leu100 on the C-helix of the same chain, and Ile28, Ile55, and Ala80 on the β-sandwich of the opposing chain (Fig 7B). The rotational nature of cNMP swing-out in the apo-form closes this gap, causing the C-helix and β-sandwich hydrophobic faces to interact directly (Fig 7B). All three of these hydrophobic features are conserved in Bd1971 homologues in other *Bdellovibrio* relatives.

In various non-CRP cNMP sensor proteins (such as the cNMP-gated potassium channel MlotK1 from the bacterium *Mesorhizobium loti*; Clayton *et al*, 2004), a helical subdomain (the N3A motif) packs against the β-sandwich and linker helix of the cNMP fold, and communicates conformational changes to the different output domains. This N3A motif projects several bulky hydrophobic residues toward the cNMP body and undergoes remodeling in the absence of ligand. Superimposition of MlotK1 with Bd1971 (via cNMP β-sandwich) shows that linker helix hydrophobic residues Ala140/Leu141/Ile144 positionally match those from the N3A motif, despite these regions having a different architecture (Bd1971 single

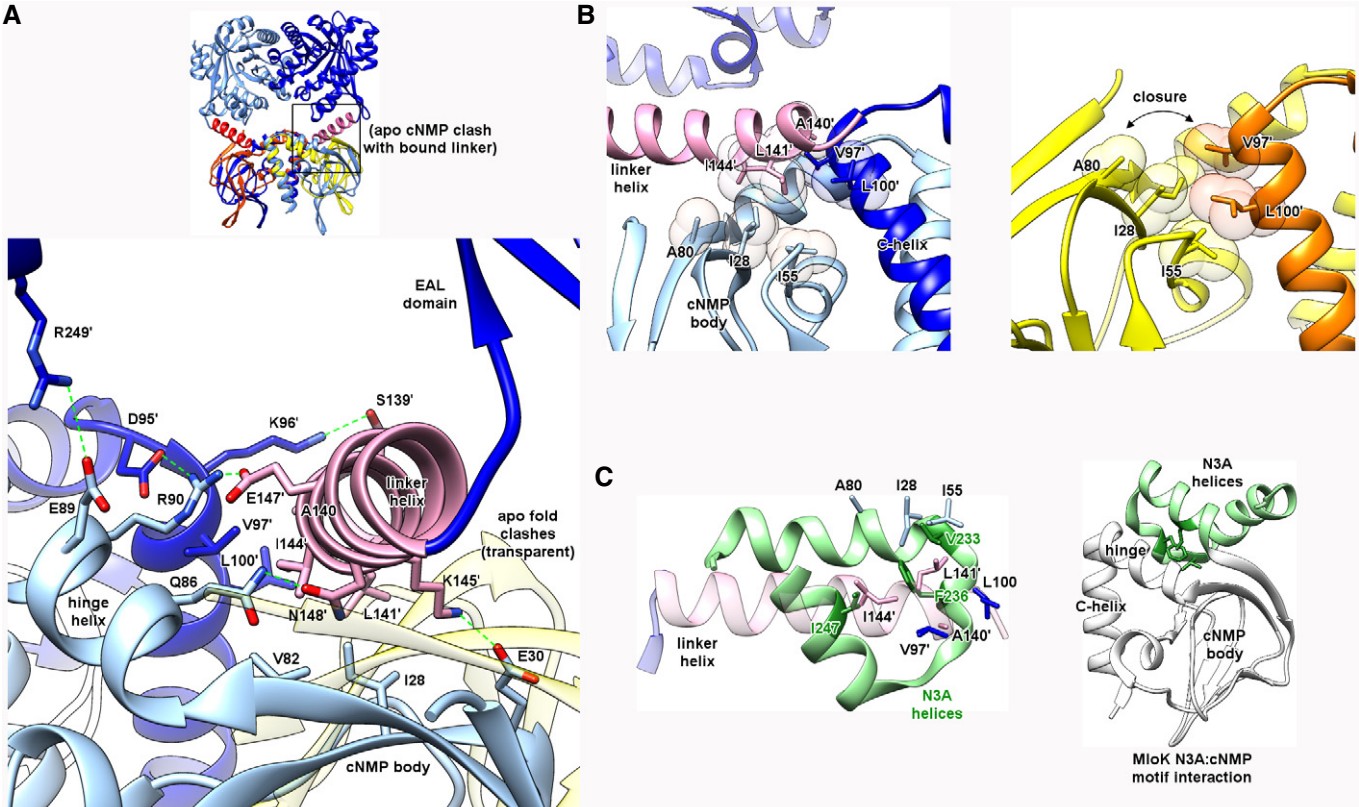

**Figure 7. The Bd1971 linker domain and relevance to signaling.**

Interactions of the linker helix (pink) with the cNMP sensor domain (dark/light blue in cAMP-bound form, yellow/orange in apo state, residues from the opposing monomer labeled by a').

A Side chains along the full span of the linker helix contact those from both chains of the sensor dimer, forming both polar (S139, K145, E147, N148) and hydrophobic (A140, L141, I144) interfaces. The linker helix conformation in the bound state would be sterically incompatible with the position of the cNMP body in the apo state (transparent, overlaid).

B The hydrophobic A140/L141/I144 triplet forms a "wedge" that inserts between two faces formed by the cNMP body (I28, I55, A80) and C-helix (V97', L100'). In the apo state, this linker feature is absent, allowing direct contact of these faces (closure of cleft).

C Analogous role of N3A motif (green) from the cNMP sensor of *M. loti* MloK (PDB 1vp6, main fold in white). The hydrophobic patch in Bd1971 linker helix formed by A140/L141/I144 is spatially equivalent to one from MloK (V233/F236/I247) despite a difference in secondary structure between the two proteins. Similarly to Bd1971, the MloK N3A hydrophobic patch contacts the fold at a region between the hinge and cNMP body (right).

helix, N3A of MlotiK1 a 3-helix bundle, Fig 7C; Clayton *et al*, 2004). The C-terminal end of the Bd1971 linker helix superimposes well with small helical elements that precede the catalytic domain of other EAL structures, including PdeL (Sundriyal *et al*, 2014; also known as YahA). Taken together, these observations provide a strong indication that the different conformations of the hydrophobic residues observed in the apo- and bound state have a functional relevance to conformational signaling.

### Differential co-operativity associated with the binding of Bd1971 sensor domains to cyclic nucleotide stimuli

Differential or selective binding of cAMP and cGMP has been observed in several cNMP domain proteins (Gorshkova *et al*, 1995), and the results of our PNPP assays indicate that cAMP and cGMP differentially stimulate Bd1971 phosphodiesterase activity. The sensor domains of the CRP protein of *E. coli* are able to bind both cAMP and cGMP, and the thermodynamics

associated with these interactions indicate that, whilst there is no co-operativity with respect to cGMP, cAMP is bound with negative co-operativity (Gorshkova *et al*, 1995). In this paradigm, isothermic CRP:cGMP binding data fitted to a two-site model indicates that binding to both sites is enthalpically favorable and that the associated heat changes for each binding event are equivalent. In contrast with this observation for cGMP binding, CRP-cAMP-binding provides a distinctive biphasic isotherm; initial binding is enthalpy driven, whilst secondary binding is enthalpically unfavorable and instead driven by an increase in entropy. In contrast, the cNMP domains of the MlotiK1 cNMP-responsive $K^+$ channel bind cAMP or cGMP non-co-operatively (Cukkemane *et al*, 2007).

We conducted a series of isothermal calorimetry experiments to explore whether binding of Bd1971 to cAMP or cGMP is subject to co-operativity and whether either nucleotide is bound preferentially. Full-length apo-Bd1971 readily aggregated when exposed to prolonged periods at 25°C at the concentrations necessary for

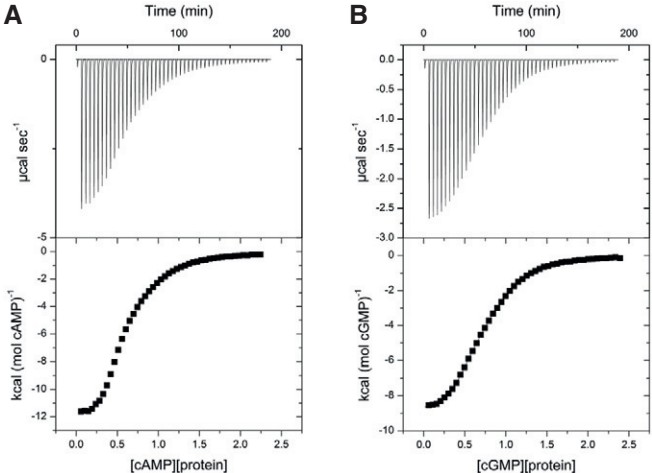

**Figure 8. Binding of cAMP and cGMP to the cNMP domain of Bd1971 measured by isothermal calorimetry.**

A, B  ITC titration curves and relative heats of binding for (A) cAMP and (B) cGMP binding to Bd1971ΔEAL. Successive injections of 6 μl of cNMP (0.5–1.6 mM) were made into 1.4 ml of apo-Bd1971ΔEAL (50–150 μM) and heat changes measured. The top panels show the ITC titrations, and the bottom panels show the binding isotherm associated with each ITC experiment. Data were analyzed with Origin software, and thermodynamic parameters associated with these binding experiments are presented in Table 2. Results presented are representative of two independent experiments.

calorimetry experiments, so apo-Bd1971ΔEAL was utilized in these experiments. Sigmoidal isotherms were obtained from titrations of Bd1971ΔEAL with cAMP (Fig 8A) or cGMP (Fig 8B). The ITC data were fitted to a two-site binding model and the resulting thermodynamic parameters obtained (Table 2).

Consistent with our crystallographic data, the calorimetry experiments indicate that each Bd1971ΔEAL dimer binds 2 molecules of cAMP or cGMP. The dissociation constants for the binding of cGMP by Bd1971ΔEAL were determined as 11.38 μM and 0.769 μM for the first and second binding sites, respectively, indicating positive co-operativity. Contrastingly, dissociation constants of cAMP for Bd1971ΔEAL were determined as 0.775 μM and 22.4 μM, indicative of negative co-operativity. The initial binding of cAMP is accompanied by a large and favorable enthalpy change but also a large loss

**Table 2. Parameters determined from isothermal calorimetry experiments.**

|  | cAMP | cGMP |
|---|---|---|
| $N^1$ | $0.432 \pm .0043$ | $0.508 \pm .0046$ |
| $N^2$ | $0.456 \pm .0098$ | $0.480 \pm .0095$ |
| $K^1_{ass}$ ($10^4$ $M^{-1}$) | $129 \pm 6.22$ | $8.79 \pm 2.09$ |
| $K^2_{ass}$ ($10^4$ $M^{-1}$) | $4.46 \pm .183$ | $130 \pm .183$ |
| $\Delta H^1$ (cal/mol) | $-12070 \pm 34.42$ | $-5203 \pm 218$ |
| $\Delta H^2$ (cal/mol) | $-7139 \pm 302.9$ | $-9023 \pm 39.9$ |
| $\Delta S^1$ (cal/Kmol) | $-12.50$ | $5.18$ |
| $\Delta S^2$ (cal/Kmol) | $-2.66$ | $-2.28$ |

of entropy. The second binding of cAMP is characterized by favorable but more modest changes in enthalpy and entropy. We reason that the differing co-operativity observed is driven primarily by enthalpy with entropy having a lesser effect since the dissociation constants for each binding event roughly correlate with the associated $\Delta H$ values. Whilst the $K_{diss}$ values for high-affinity Bd1971 binding of cAMP and cGMP were equivalent (0.775 μM for cAMP, 0.769 μM for cGMP), it was noted that the $K_{diss}$ for low-affinity cAMP-binding was higher than that for cGMP. In total, Bd1971ΔEAL binds the likely physiological ligand cAMP with negative co-operativity and with lower affinity than cGMP, which is bound with positive co-operativity.

### Differences in cNMP co-operativity are reflected by changes in Bd1971 phosphodiesterase activity

To determine how the co-operative binding of cAMP and cGMP by Bd1971 effects phosphodiesterase activity, magnesium-supplemented apo-Bd1971 was titrated with cNMPs and hydrolysis of PNPP was assayed (Fig 9). The kinetic data were fitted to an allosteric sigmoidal model (this model provided a better fit than other models and is in agreement with the calorimetry and structural data) and the associated kinetic parameters determined. Bd1971 phosphodiesterase activity increased with cAMP or cGMP concentrations until saturation. As in Fig 5, Bd1971 saturated with cGMP had a higher phosphodiesterase activity than when saturated with cAMP. Binding of cGMP also gave a lower $K_{half}$ value than cAMP. Intriguingly, Hill coefficients of 1.85 and 1.94 were calculated for cAMP and cGMP, respectively.

We interpreted these observations with reference to the ITC data to determine whether the phosphodiesterase activity of dimeric Bd1971 is stimulated by binding of one or two molecules of cAMP or cGMP: *i.e.,* asking whether both cNMP domains need to be occupied for catalytic output by the EAL domains or whether these domains function independently. Three key points indicate that both cNMP domains must be occupied for the activation of Bd1971 phosphodiesterase activity. First, activity is stimulated by cAMP at concentrations higher than cGMP. If activity was stimulated by binding to a single cNMP domain, then one would expect cAMP, which is bound by Bd1971 with high affinity initially and then lower affinity (as determined by ITC), to activate Bd1971 progressively and at lower concentrations than cGMP. Second, the $K_{half}$ values of Bd1971 phosphodiesterase activity recorded in the presence of cAMP and cGMP are approximately twofold higher than the $K_{diss}$ values of associated with their respective low-affinity binding sites determined from ITC experiments (respective $K_{half}$ and $K_{diss}$ values of 54.22 and 22.4 μM for cAMP, 26.17 and 11.38 μM for cGMP). This observation is consistent with the proposal that occupation of the low-affinity cNMP site is the limiting step in Bd1971 activation. Since cGMP is bound with positive co-operativity (the $K_{diss}$ for the second binding of cGMP is 0.769 μM), binding of cGMP to the second, high-affinity site, of Bd1971 can be regarded as occurring simultaneously with binding to the first, low-affinity site. Third, the Hill coefficient of 1.85 for cAMP is contrary to expectations from the ITC analysis. If phosphodiesterase activity was stimulated by binding of cAMP to a single Bd1971 cNMP domain, then we would expect a Hill coefficient of < 1, consistent with the negative co-operativity for this ligand recorded by ITC. However, if both cNMP

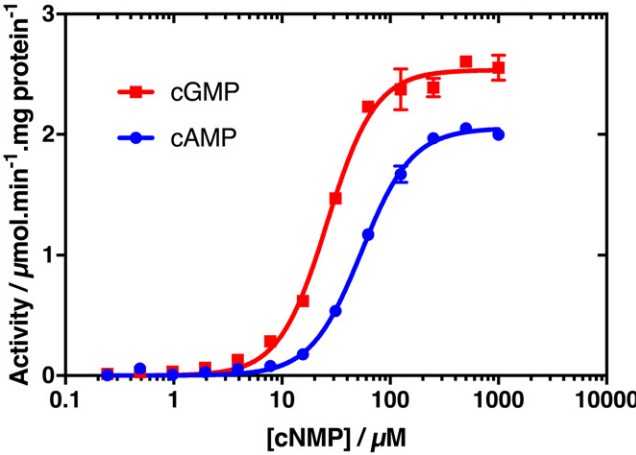

**Figure 9. Phosphodiesterase activity of Bd1971 in relation to co-operativity of cNMP ligand binding.**

Data were fitted with an allosteric sigmoidal model in GraphPad Prism software and used to determine the kinetic parameters associated with cNMP-dependent phosphodiesterase activity. Data points for both sets of results are the average of three experimental replicates with the standard deviation shown by intersecting bars. Results are representative of at least three independent experiments.

Source data are available online for this figure.

domains must bind cAMP for Bd1971 to become catalytically active, then the initial binding of cAMP would not register in our activity assays; hence, we extract Hill coefficients indicative of positive co-operativity for both the positive c-operative ligand, cGMP, and the negative co-operative ligand, cAMP. To summarize, we propose that Bd1971 EAL domains are competent for phosphodiesterase activity only when both cNMP domains are occupied with cAMP or cGMP.

**The structure of a half-occupied sensor dimer identifies features that promote a stable, intermediary conformational state**

The negative co-operativity of the Bd1971$^{\Delta EAL}$ cNMP sensor dimer raised the prospect of a potentially stable (intermediary) state in which the protein could be half-occupied by a single cAMP—this is important given that there is little to no structural information on any intermediary state for other cNMP sensors. In screening for cNMP-bound Bd1971$^{\Delta EAL}$, lower concentrations of cAMP did indeed result in different crystal forms, and we were able to solve the structure of a half-occupied dimer (hereafter referred to as Bd1971$^{\Delta EAL-mid}$) to 2.79 Å resolution. This form has two dimers in the asymmetric unit, each binding a singular cAMP molecule, the fold traceable until the end of the C-helix (residue 114). Strikingly, the conformational cNMP swing-out between the apo and bound forms is replicated to an intermediary degree in the cAMP-occupied chain of Bd1971$^{\Delta EAL-mid}$, with a rotation of ~30° and translation (at β-hairpin) of 10 Å (Fig 10A, aligned via C-helices which again do not shift relative to one another). Significantly, the empty sensor chain of Bd1971$^{\Delta EAL-mid}$ adopts the "fully outward" domain juxtaposition observed in both chains of the apo-form. Our analysis of the elements driving conformational change between bound full-length Bd1971 and apo Bd1971$^{\Delta EAL}$ is further supported by adding the Bd1971$^{\Delta EAL-mid}$ occupied chain to the ensemble; the C-

helix:cNMP body swivel is only partly engaged (Fig 10B, aligned via cNMP fold) and the P-loop stabilization involving the movement of Arg110 to contact Asp63 is at an intermediary position (Fig 10C). Using the distance between the C$_\alpha$ atoms of hinge residue Asp95 and small helix Ile62 as a measure of P-loop migration, the Bd1971$^{\Delta EAL-mid}$ structure (16.7 Å) sits between that of the apo-form (20.1 Å, swung out) and two-site bound state (14.6 Å, swung in).

The nucleotide-bound pocket of Bd1971$^{\Delta EAL-mid}$ reveals cAMP to be bound in the syn conformation; of the two full-length, bound-state structures of Bd1971, this pose resembles the cGMP complex (syn) more closely than the cAMP complex (anti, Fig 10D). The syn nucleotide complexes place the Hoogsteen N6 & N7/O7 edge toward solvent and the Watson-Crick N1 edge toward the C-helix of the same chain, whereas the anti-cAMP complex reverses this arrangement. However, both the anti-cAMP and syn cGMP conformations allow for the side chain of Met59 to occupy the "down" rotamer that stacks with the nucleotide base, whereas the syn cAMP Bd1971$^{\Delta EAL-mid}$ conformation places Met59 in a conformation more similar to the full-length apo structure (Fig 10B and D). The result of this difference between the different forms is that the full-length complexes position the neighboring Arg108 residue (corresponding to the conserved hydrophobic cap in other cNMP sensors; Kornev et al, 2008) in an orientation that promotes full cNMP body:C-helix closure (2.6–2.9 Å from the carbonyl of Ile55), whereas the Bd1971$^{\Delta EAL-mid}$ syn cAMP complex results in a kinked conformation of the Arg108 cap (5.15 Å away from Ile55), resulting in the partial domain closure observed (Fig 10B and D).

**Bacterial two hybrid assays reveal interactions of Bd1971 with *Bdellovibrio* c-di-GMP synthase family proteins**

Previously, it has been suggested that asymmetry in CRP, triggered by single site binding of cAMP, could be important in mediating associations and dissociations with target DNA sequences (Popovych et al, 2006). However, our phosphodiesterase activity assays indicate that both Bd1971 cNMP sites must be occupied for catalysis, rendering any functionality of the transition state obscure. We reasoned that the two-step conformational change of Bd1971 upon binding cAMP could have an effect on protein–protein interactions. EAL domain proteins have been shown to interact with GGDEF proteins in other bacteria (Sarenko et al, 2017), so we tested for such interactions with Bd1971 by bacterial two hybrid (BTH) assays, expressing *B. bacteriovorus* GGDEF proteins. Interactions were demonstrated (Fig 11) with both Bd0367 (DgcA; a GGDEF protein, previously shown to be important for gliding motility) and Bd3125 (CdgA; a degenerate GGDEF protein involved in prey entry by the *Bdellovibrio*; Hobley et al, 2012). We further tested which domain of Bd1971 was interacting with Bd3125 by repeating the BTH experiments with either the N-terminal cNMP domain (here by cloning residues 1–145) or the C-terminus EAL domain (residues 145-400). Whilst there was some interaction with the C-terminus, the N-terminus interacted significantly more (Fig 11A and B). Curiously, we only found an interaction with the response regulator domain (residues 1–139) of Bd0367, and not the full-length protein, suggesting that the full-length protein may interfere with the BTH. Testing interactions between Bd1971 and DgcB (Bd0742) by BTH showed a

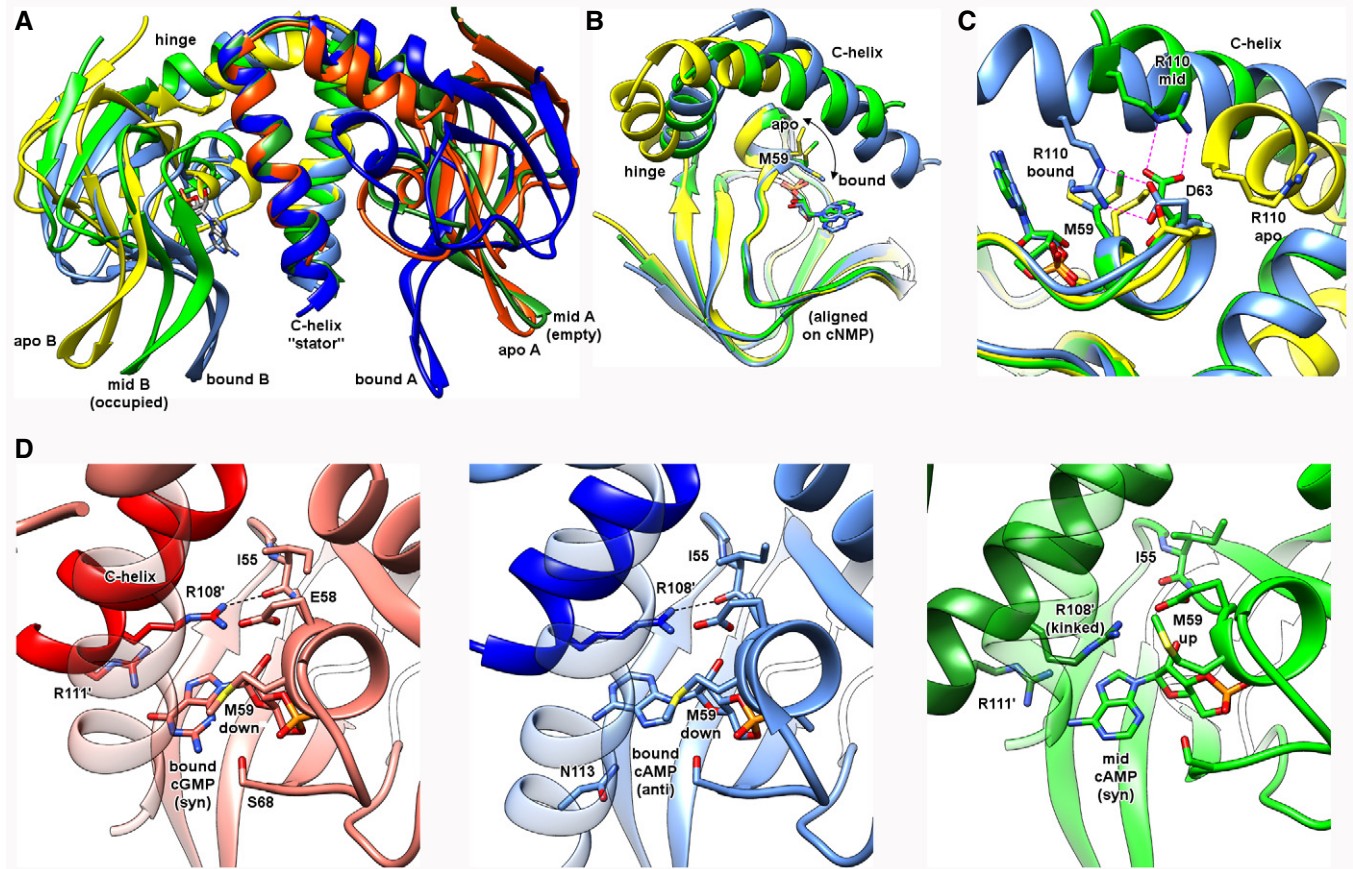

**Figure 10.  Isolation and observation of an intermediate conformational state of Bd1971, relating to a half-occupied sensor dimer.**

Different forms colored thus—cAMP-bound full-length (dark/light blue), cGMP-bound full-length (red), apo Bd1971$^{\Delta EAL}$ (yellow/orange), and half-occupied cAMP Bd1971$^{\Delta EAL\text{-}mid}$ (dark/light green). Residues from the opposing monomer labeled by a'.

A    Comparative conformational shifts when aligned on C-helix dimer. One monomer from the half-occupied form is empty (mid A) and adopts a similar pose to the apo-form; the other monomer has bound cAMP (mid B) and adopts a pose halfway between the apo and bound forms.

B, C  The occupied cNMP monomer of Bd1971$^{\Delta EAL\text{-}mid}$ structure (green) possesses other features conformationally intermediate between the apo (yellow) and bound (blue) states, namely the position of M59 and the juxtaposition of the D63:R110 pair.

D    Comparison of nucleotide conformations between the three differing bound states of the Bd1971 cNMP sensor. The cAMP ligand in the Bd1971$^{\Delta EAL\text{-}mid}$ structure (green) is present in the syn form, with nucleotide base pointing toward the cyclic ribose phosphate. This conformation is like that observed for cGMP (red), but opposite to that of cAMP in the full-length (blue, occupied at both sites) complex. The interaction of R111′ with the syn cAMP in Bd1971$^{\Delta EAL\text{-}mid}$ kinks R108 such that it cannot make a contact with the backbone carbonyl of I55, leaving M59 in upward conformation.

significant interaction only for proteins expressed in one combination of BTH plasmids (Fig EV2) suggesting a weak or transient interaction. Testing interactions between Bd1971 and DgcC (Bd1434), a protein important for non-predatory growth, did not show significant interactions (Fig EV2).

## Bd1971 mutants do not exhibit associated phenotypes of mutants of the interacting c-di-GMP proteins

Each c-di-GMP synthetic/binding GGDEF protein of *Bdellovibrio* was previously shown by gene deletion studies to be linked to a discrete function (Hobley *et al*, 2012), so we investigated the possibility that Bd1971 was involved in phenotypes related to (or, where possible, the opposite of) those of any of its interacting c-di-GMP synthase partners. The rationale was that deletion of Bd1971, which we have shown herein depletes intracellular c-di-GMP levels, might lead to

an exaggerated or aberrant phenotype associated with the activities of one or more GGDEF proteins. Previously, mutants of CdgA (Bd3125) exhibited slow growth, unable to clear lawns of prey cells on double-layer overlay plates and taking 2 days to clear prey in liquid cultures, whilst mutants of DgcB (Bd0742) were completely unable to enter prey and could only grow host-independently (Hobley *et al*, 2012). The ΔBd1971 strain did not exhibit improvements above wild type, in plaque forming, or predation phenotypes as it readily formed plaques on double-layer overlay plates and cleared prey from liquid cultures overnight as did the wild-type strain HD100.

Mutants of DgcA (Bd0367) can enter and kill prey cells, replicate within, and lyse the prey, but cannot glide out when immobilized on a surface (Hobley *et al*, 2012). We therefore tested the ΔBd1971 strain for gliding ability by immobilizing on 1% agarose pads and observing by time-lapse microscopy. Similar to wild type, the

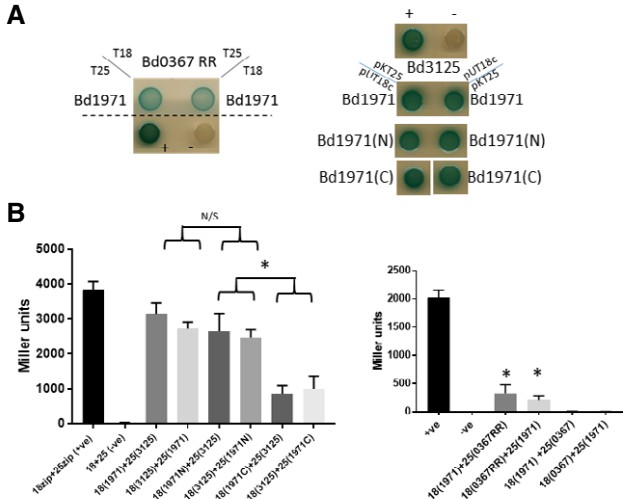

**Figure 11. Bacterial two hybrid (BTH) assays showing interactions between Bd1971 and c-di-GMP synthesizing or binding proteins.**

A  Spot tests of co-transformants of either Bd0367 (DgcA; GGDEF protein) response regulator domain or Bd3125 (CdgA; a degenerate GGDEF protein likely involved in c-di-GMP binding and signaling) with full-length Bd1971, or the N-terminal (1–145), or C-terminal (145–400) portions of Bd1971 in either pUT18c or pKT25 as indicated. Positive control was co-transformants with pUT18c-zip and pKT25-zip; negative control was the empty vectors. The co-transformants were plated on LB-X-Gal medium with positive interaction resulting in blue coloration, which can be seen for all of the interactions in both plasmid combinations, but not for the negative control. Images are representative of at least 4 independent experiments.

B  Confirmation of BTH interactions by β-Galactosidase activity assays. The interaction is significantly stronger between Bd3125 and the N-terminal domain of Bd1971 than the interaction with the C-terminus of Bd1971. The response regulator domain of Bd0367 showed significant interaction with Bd1971, but the full-length Bd0367 did not. *$P > 0.05$ by Student's $t$-test N/S—not significant. Data are from at least three independent experiments; error bars are the standard error of the mean

Source data are available online for this figure.

majority of the population of cells were observed to glide, with no significant delay in the onset of gliding (Fig EV3), showing that this mutation does not affect, or enhance, gliding motility.

## Discussion

In total, our results illustrate the specific mechanism by which a cAMP sensory domain can regulate a fused EAL domain for the purposes of c-di-GMP hydrolysis. Our results provide one of the first structural/mechanistic investigations into linkage between cAMP and c-di-GMP in bacteria. Proteins with a similar cNMP/EAL domain architecture also exist in other predatory bacteria taxonomically distant to δ proteobacterium *Bdellovibrio* (e.g., the α-proteobacterial epibiotic predator *Micavibrio aeruginosavorus*, or the γ-proteobacterial wolf pack predators of *Lysobacter* spp.). Beyond predators, cNMP/EAL full-length homologues of Bd1971 are of wider interest as they can be found in multiple proteobacteria and cyanobacteria, selected spirochetes and actinobacteria, but have not been studied structurally previously (Fig EV4). Hence, Bd1971 and

homologues have the potential to directly integrate signaling from both cAMP and c-di-GMP pathways, distinct to the indirect means employed by *Vibrio cholerae* (cAMP repressing GGDEF expression via a CRP equivalent) (Fong & Yildiz, 2008) and *Pseudomonas aeruginosa* (surface sensing resulting in production of both nucleotides) (Luo *et al*, 2015). Also the cAMP stimulation of EAL activity in *Bdellovibrio* is in contrast to that of GGDEF activation observed in *Leptospira* (which senses cAMP via a GAF domain in GGDEF protein Lcd1; da Costa Vasconcelos *et al*, 2017).

The full-length structure of cAMP-activated Bd1971 provides a means for understanding EAL regulation in a multidomain protein wherein the sensory domain is appended at the catalytic end of the EAL barrel; the well-characterized RocR and BlrP1 EAL enzymes regulate dimerization/activity from the opposing end of the fold (Barends *et al*, 2009; Chen *et al*, 2012). Comparison of Bd1971, RocR, and BlrP1 reinforces the importance of classical EAL dimerization for phosphodiesterase activity, but suggests that there are different means by which dimerization/activity may be controlled. The RocR and BlrP1 domain arrangements, whilst different to one another, both place sensory domains against the EAL dimerization helix and thus are able to directly couple sensor activation to enzymatic activity. The Bd1971 domain arrangement places the sensor far from this region (40 Å) and thus importantly provides the first observation that demonstrates EAL activity can be regulated at a distance by large-scale domain movements.

The Bd1971 cNMP sensor domain exhibits similar dimerization to the cognate domain in the CRP family of proteins, but uses a different connection to the central C-helix, "swapping over" into the opposing monomer. It may now be possible to predict whether other cNMP-containing proteins adopt a Bd1971-like, crossover distinct from the "usual" CRP arrangement—constraints derived from the resolved structures and a sequence alignment of diverse homologues suggest that crossover requires a small residue (A, G, S, or T) at a position equivalent to Ala94.

The signaling mechanism of cNMP superfamily members has previously been broadly categorized into two archetypes: cis-acting, often monomeric, sensors (e.g., those found in protein kinases) that shift the C-helix relative to the β-sandwich body, or trans-acting homodimers (e.g., like CRP that signal across the domain interface to alter C-helix properties) (Kornev *et al*, 2008). The structure of Bd1971 now defines a new, third distinct signaling type—one that creates a cNMP domain "swing-out" from a central axis—this is achieved by melding the C-helix movement of the cis grouping to the dimerization "stator" of the trans-grouping. The N3A motif of the cis grouping that communicates the signaling-associated conformational change to the output domain is replaced in Bd1971 by a spatially equivalent motif formed by the linker helix. The absence of any linker helix region density in both our apo- and half-occupied structures (determined using the Bd1971$^{\Delta EAL}$ construct) is suggestive of frequent/rapid linker remodeling, as there is no space in either structure to accommodate residues Ala140/Leu141/Ile144 in the orientation that they adopt in the full-length mononucleotide bound state. Hence, at least some rearrangement is necessary when the cNMP domain swings upward to occlude this region. Observation of both a half- and fully swung out state, and adherence to the P-loop-driven mechanisms used by other disparate sensors support the idea that the associated conformational changes are not artefacts of this construct lacking the EAL domain.

The similarity in binding pose between an anti-cAMP and syn cGMP has been observed in some other cNMP sensor proteins, including CRP, where the cAMP N6 and cGMP O6 occupy similar positions (Passner *et al*, 2000; Seok *et al*, 2014). This binding mirrors the dominant forms present in solution, with a syn: anti ratio for cAMP of 30:70 and cGMP of 95:5 (Yathindra & Sundaralingam, 1974).

The activation of Bd1971 by both cAMP and cGMP is in contrast to CRP, wherein cGMP is unable to trigger activation via C-helix elongation (and acts to lock CRP in an inhibited state; Seok *et al*, 2014); this difference is easily explained by the different mode of Bd1971 C-helix usage outlined above. The other notable difference between the Bd1971 anti-cAMP and syn GMP complexes is that Ser68 can make a specific interaction to N2 of the guanine ring (Fig 10D)—protein kinases specific for cGMP often encode a Ser/Thr at this position in contrast to cAMP-specific domains utilizing Ala (Osborne *et al*, 2011). It is not immediately apparent why Bd1971 and various CRP proteins activated by cAMP use Ser at this position, although this may be related to its secondary role in contacting the nucleotide phosphate oxygen. A Ser to Ala mutant at this position in *E. coli* CRP loses the ability to bind cGMP (Seok *et al*, 2014). It was thus hypothesized that cGMP inhibition of CRP acts to avoid triggering activation when adenylate cyclases erroneously produce cGMP; in this regard, Bd1971 phosphodiesterase activity would be stimulated whether "correct" cAMP or "incorrect" cGMP was produced. *Bdellovibrio* is not known to produce cGMP, unlike other δ-proteobacteria such as Myxobacteria (Gomelsky & Galperin, 2013). Production of cAMP is likely to arise at different phases in the *Bdellovibrio* life cycle as a product of one or several of the annotated adenylate cyclases encoded from the genome of strain HD100 (*bd0081, bd1116, bd2640, bd3072*—subject of an ongoing study beyond the scope of this work). Identifying possible phosphodiesterases involved specifically in cessation of cAMP signaling is less straightforward due to the high density of hydrolytic enzymes encoded by *Bdellovibrio*.

The isothermal calorimetry results indicate a clear negative co-operativity for cAMP-binding to the Bd1971 sensor dimer, and positive co-operativity for cGMP. The $K_{diss}$ values of 0.8 and 22.4 µM for the two Bd1971 cAMP-binding events display equivalent relationships to those measured for trans-acting CRP members (0.04/4 µM for *E. coli*, 17/130 µM for *C. glutamicum* proteins, respectively; Gorshkova *et al*, 1995; Townsend *et al*, 2014). Hence, despite a different C-helix:β-sandwich interface, Bd1971 retains the ability of trans-acting proteins to signal occupancy status between the two binding sites. NMR and computational modeling support a dynamics-driven model for *E. coli* CRP negative co-operativity, where binding of the first molecule of cAMP acts to alter protein motions rather than induce a structural change at the second binding site (Rodgers *et al*, 2013). To the best of our knowledge, our structure determination of Bd1971$^{\Delta EAL\text{-}mid}$, with syn cAMP bound at one chain (and the other chain empty), is the first example of a dimeric cNMP sensor trapped in a singly occupied state (the 2006 study of Popovych *et al* on half-occupied CRP focused on comparative dynamics as opposed to discrete structures; Popovych *et al*, 2006).

Consistent with our kinetic data, full-length Bd1971 binding cAMP at a single site is inactive because both the fully swung out empty cNMP domain and half swung out occupied domain clash with the productive linker helix position—both chains have to adopt the swung in conformation to allow classical EAL dimer formation. The Bd1971$^{\Delta EAL\text{-}mid}$ conformation we observe is informative about how allostery is communicated through the cNMP sensors, implicating two separate events upon nucleotide binding. The syn cAMP in Bd1971$^{\Delta EAL\text{-}mid}$ has triggered the primary event (P-loop movement as it engages with nucleotide ribose phosphate), without triggering the secondary event (the lack of correct hydrophobic cap formation over the base, involving Arg108 and neighboring residues, that ultimately acts to close the β-sandwich against the C-helix). A two state mechanism is supported by studies on protein kinase G, indicating that cNMP domains can trigger P-loop/C-helix events independently of one another (Osborne *et al*, 2011); furthermore, the cGMP-bound form of *E. coli* CRP demonstrates P-loop engagement without conventional hydrophobic capping (Seok *et al*, 2014). The CRP part-triggered pathway shares similarities to Bd1971—residues L124 and L73 of CRP do not stack in the same manner as the conventionally bound state and have direct equivalents in Bd1971 (L109 and M59, respectively) that change their stacking interaction with base between the part-triggered and fully-triggered states (Fig 10). The empty site of Bd1971$^{\Delta EAL\text{-}mid}$ is different to those observed in the Bd1971$^{\Delta EAL}$ apoprotein (in gross conformation at least) and so will present a different cleft to incoming cAMP—this may be responsible for the observed negative co-operativity, or like CRP (Rodgers *et al*, 2013), it may arise from different dynamics relating to the (changed) pose of the occupied monomer (Fig 10A). The change from a singly occupied to a doubly occupied state we predict will be accompanied by a syn to trans conversion of bound nucleotide.

The relative instability of apo full-length Bd1971 prevents observation of what happens to the EAL domain upon loss of cAMP at the N-terminal sensory domains. The cNMP swing-out we observe in two states of the Bd1971$^{\Delta EAL}$ construct (and similarity to motions of structurally related cis-acting superfamily members) indicates that linker domain remodeling in the apo-form will cause some form of structural change to the associated EAL domain. There are two possibilities that could result in the required cessation in enzymatic activity—either the EAL active site will become occluded (similar in mechanism to RocR; Chen *et al*, 2012), or the EAL domains will become forced into a non-productive orientation (similar in mechanism to most structurally characterized EAL enzymes). The relatively short distance between the linker helix and Bd1971 c-di-GMP binding site (15 Å) does not rule out the first mechanism, and it is interesting that the active site "wing" between these points (182–192) adopts a different conformation between chains A of the full-length structure (helical in chain A, β-hairpin in chain B), indicating inherent malleability. The second mechanism may be probable, given the propensity for EAL domains to adopt differing dimeric states.

Our results prompt a consideration as to what features may be observed in a full-length apo-form of Bd1971. A comparison of Bd1971 to the (non-cNMP-binding) *E. coli* c-di-GMP phosphodiesterase PdeL is particularly informative—PdeL has been captured in both closed and offset open dimers (Sundriyal *et al*, 2014), and these states appear to be physiologically relevant because they have also been observed in other EAL protein structures that utilize a completely different means of regulation. Furthermore, mutations of PdeL that disrupt regulation appear to affect the conformational equilibrium between these different dimeric forms (Sundriyal *et al*,

2014). The N-terminal helix of PdeL (residues 110–120) is structurally identical to the C-terminal half of the Bd1971 linker helix (residues 147–157), but the residues preceding this in PdeL (104–109) are in a β-conformation in the closed dimer (PDB 4LJ3) and a distorted conformation in the open dimer (4LYK) (Sundriyal *et al*, 2014). Hence, the agreement of the N-terminal half of the Bd1971 linker helix (residues 136-146) with the N3A motif that undergoes remodeling in other cNMP sensor proteins (Fig 7c) could provide a means to destabilize the helical conformation at the start of the linker helix and thus switch Bd1971 into a different state (akin to PdeL closed, inactive) upon loss of cAMP ligand.

The resolved structures confirm that the Bd1971 EAL is inactivated via prevention of classical dimer formation (this being incompatible with the observed location of the apo-form cNMP domains). Cumulatively, the biological data showing that *bd1971* is constitutively expressed, that Bd1971 is present throughout the cell body, and its absence results in higher global c-di-GMP levels suggest that its role is one of regulating these levels throughout the *Bdellovibrio* cell. Further, its interaction with several GGDEF proteins hints at a cellular role for Bd1971 binding and possibly inhibiting c-di-GMP production, in addition to direct degradation. The nature of EAL/GGDEF activity within a protein "hub" can be extremely complex (particularly +/− stimuli or inhibition; Lindenberg *et al*, 2013) and is beyond the scope of this study; regardless one can see a requirement for "action potential" spikes of c-di-GMP to be produced during key timepoints of the staged *Bdellovibrio* life cycle but may be too transient to assay phenotypically. Non-disturbance of both invasion and gliding in the Δ*bd1971* strain may be indicative that excess c-di-GMP is tolerated in these short-term assays, but would be wasteful (too extended a period of stimulation), and evolutionarily selected against, in the long term.

Our observation of cAMP stimulation of Bd1971 activity encourages our further analysis of cAMP signaling during predation. Herein, it is interesting to speculate on the nature of cAMP/c-di-GMP overlap; if cAMP reports on the energy state of the predator, levels would potentially be at their highest pre-invasion (starvation); this may tally with a need to synchronize Bd1971 with DgcB and CdgA, which are utilized during and after this timepoint (in keeping with results from the interaction assays). Suppression of the GGDEF-mediated switches (DgcA gliding, DgcB invasion) could be a useful means by which *Bdellovibrio* could efficiently use its finite resources, e.g., whilst growing within the bdelloplast, mechanisms for stalling prey exit until resources are exhausted could be beneficial. Conversely, outside of the bdelloplast, activating the potential to enter prey, glide toward prey or grow host-independently would be favorable. It is clear from our work that a Bd1971-catalyzed rapid activation/deactivation of global c-di-GMP hydrolysis is possible (via this novel cNMP swiveling control of the EAL domain) pertinent to the rapidly changing nutritional state of the *Bdellovibrio* lifestyle (from prey hunting to intrabacterial to environmental).

# Materials and Methods

### RNA isolation from the predatory cycle and RT–PCR analysis

Synchronous predatory infections of *B. bacteriovorus* HD100 on *E. coli* S17-1 prey in Ca/HEPES buffer (2 mM CaCl$_2$ 25 mM HEPES pH 7.6), or strain S17-1 suspended in Ca/HEPES alone, were set up as previously described (Lambert *et al*, 2006) with samples throughout the timecourse being taken and total RNA isolated from them. This semi-quantitative PCR allows the evaluation of specific predator transcripts in the presence of fluctuating levels of prey RNA as the predator degrades it. RNA was isolated from the samples using a Promega SV total RNA isolation kit with the RNA quality being verified by an Agilent Bioanalyser using the RNA Nano kit. PCR was performed on each RNA sample in the absence of reverse transcriptase to confirm that there was no contaminating DNA. RT–PCR was performed with the Qiagen One-step RT–PCR kit with the following reaction conditions: one cycle 50°C for 30 min, 95°C for 15 min, then 25 cycles of 94°C for 1 min, 50°C for 1 min, 72°C for 1 min, a 10-min extension at 72°C after the 25 cycles, and finally a 4°C hold. Two independent repeats were carried out. Primers to anneal to *bd1971* were 5′- GATGGTGATTCCGATTGGTC -3′ and 5′- TTCAGAC GCAACTGATCCTG -3′; primers to anneal to *dnaK* were 5′- TGAGGA CGAGATCAAACGTG -3′ and 5′- AAACCAGGTTGTCGAGGTTG -3′.

### Fluorescent tagging of Bd1971

The *bd1971* gene lacking its stop codon was cloned into the conjugable vector pK18*mobsacB* in such a way as to fuse the genes at the C-terminus with the mCherry gene. This fusion was introduced into *B. bacteriovorus* by conjugation as described previously (Fenton *et al*, 2010). The *bd1971* gene was amplified using the primers 5′ TT CCGAGCTCATGAATGCAGCAGCTCAGTCC 3′ and 5′ GGGGTACC GATCAGCTTTAAGAAGCGGC 3′ digested with *Sac*I and *Kpn*I and cloned into the vector pAKF56 to fuse the gene with mCherry. This was then excised by digestion with *Sac*I and *Hin*dIII and cloned into pK18*mobsacB* cut with *Sac*I and *Hin*dIII.

Epi-fluorescence microscopy was undertaken using a Nikon Eclipse E600 through a 100× objective (NA 1.25) and acquired using a Hamamatsu Orca ER Camera. Images were captured using Simple PCI software (version 6.6). An hcRED filter block (excitation: 550–600 nm; emission: 610–665 nm) was used for visualization of mCherry tag expression. We wished to determine whether any changes in localization of Bd1971 occurred during the predatory cycle as this may have implied a role in c-di-GMP reduction at a localized cellular site. For this reason, the balance and contrast levels for the fluorescence channel were individually adjusted for each timepoint image to show the mCherry as brightly as possible within the bdelloplast.

### Generating gene deletion and replacement mutants in *Bdellovibrio bacteriovorus*

Markerless deletion of *bd1971* from *B. bacteriovorus* HD100 was achieved as described previously (Lerner *et al*, 2012; Lambert & Sockett, 2013). Primers designed for *bd1971* deletion to amplify to the upstream region of *bd1971* were 5′- cgacggccagtgcca TTTTGAATACAATCTTTCAAGC -3′ and 5′-gagaacaaatATTCATATG AGTTTTATTCTAGGG -3′. Primers designed for *bd1971* deletion to amplify to the downstream region of *bd1971* were 5′- tcatatgaat ATTTGTTCTCTAAACCCGTTG -3′ and 5′- ctatgaccatgattacgTCCAC AGTTCATCACTTCC -3′. Primers designed to generate the R67D mutation were, for the upstream region, 5′- TAAAACTCATATGA ATGCAGCAGCTCAG-3′ and 5′- CACTGGCAGAAGCGTTTTGATTG -3′

and for the downstream region 5′-CAATCAAAACGCTTCTGC CAGTG-3′ and 5′-AGCTCGGTACCCGGGTTAGATCAGCTTTAAGA AGCG -3′. Primers designed to generate the D307308A mutation were, for the upstream region, 5′-TAAAACTCATATGAATGCAGC AGCTCAG -3′ and 5′-CGGTCCCGAAAGCAGCAATGGAAATTGC-3′ and for the downstream region 5′-GCAATTTCCATTGCTGCTTTCGG GACCG-3′ and 5′-AGCTCGGTACCCGGGTTAGATCAGCTTTAAGAA GCG-3′. PCR products were cloned into the pK18*mobsacB* conjugable vector by Gibson cloning using the NEBuilder kit. Mutations were confirmed by sequencing.

### Extractable c-di-GMP determinations for *Bdellovibrio* cells

The method of Bobrov (Bobrov *et al*, 2011) was used to determine the extractable levels of c-di-GMP in axenically or predatory grown *Bdellovibrio* cells with analysis of extracts using liquid chromatography tandem mass spectrometry carried out by Lijun Chen and Bev Chamberlin at the Mass Spectrometry Core of RTSF (Research Technology Support Facility) at Michigan State University USA. Host-independent cultures were back-diluted in to 10 ml PY broth supplemented with kanamycin sulfate at 50 μg/ml) to a starting $OD_{600}$ of 0.2 and grown to an $OD_{600}$ of 0.6. Predatory *Bdellovibrio* were grown in concentrated cultures of 5 ml *E. coli* S17-1 (which had grown for 16 h in YT broth with 200 rpm shaking at 37°C), inoculated with 5 ml of *Bdellovibrio* predatory prey-lysate precultures in 15 ml Ca/HEPES buffer, incubated in a Falcon tube with 200 rpm shaking at 29°C for 16 hours to allow prey lysis and release of predator cells. Eight of these were grown per strain and filtered through a 0.45-μm filter to remove any remaining prey cells. Cells were then pelleted by centrifugation, and the wet weights were determined; they were frozen in liquid nitrogen and then processed using Bobrov's method and extraction buffer (40% methanol 40% acetonitrile in 0.1 N formic acid), which was later neutralized with $NH_4HCO_3$. The levels of extractable c-di-GMP in the cell extracts were compared to known added standards of pure c-di-GMP.

### Bacterial two hybrid (BTH) assay

For bacterial two hybrid analysis of potential interactions of *B. bacteriovorus* proteins expressed in *E. coli*, each ORF was cloned in-frame with the T18 and T25 fragments of adenylate cyclase ORF in vectors pUT18/pUT18C and pKNT25/pKT25 (Karimova *et al*, 2001). The resulting vectors were then co-transformed into *E. coli* strain BTH101 and plated onto Nutrient Agar (Oxoid) supplemented with 50 μg ml$^{-1}$ Ampicillin, 25 μg ml$^{-1}$ Kanamycin, and 40 μg ml$^{-1}$ 5-bromo-4-chloro-3-indolyl-β-D-galactopyranoside (X-gal) and incubated at 29°C for 48 h. Three co-transformants for each assay were cultured to stationary phase in LB broth and spotted onto nutrient agar, supplemented as above, and incubated for 48 h at 29°C. Plates were then scanned on an Epson Perfection 1200U scanner. Beta-galactosidase activity was performed at 29°C on 1 ml stationary-phase aliquots of cultures as described by Miller (Miller, 1972).

### Measurement of gliding motility of strains with *bd1971* mutations

Cells were grown as predatory prey lysate cultures in 10 ml Ca/HEPES buffer with 600 μl *E. coli* S17-1 prey (grown for 16 h in YT broth with 200 rpm shaking at 37°C) inoculated with 200 μl of a previous prey lysate for 24 hours with 200 rpm shaking at 29°C. Ten microliter samples were immobilized on Ca/HEPES 1% agarose pads and imaged using a Nikon Ti-E inverted fluorescence microscope equipped with a Plan Apo 100x/1.45 Ph3 objective and Andor Neo sCMOS camera. Acquisition was with the Nikon NIS software. Images were analyzed using the MicrobeJ plugin for the ImageJ (FIJI distribution) software (http://www.indiana.edu/~microbej/index. html; Ducret *et al*, 2016), which automates detection of bacteria within an image. The *Bdellovibrio* cells were defined by area 0.2–1.5 μm$^2$, length 0.5–5 μm, width 0.2–1.5 μm, and all other parameters as default. Manual inspection of the analyzed images confirmed that the vast majority of cells were correctly detected. For gliding analysis, MicrobeJ was used as above to count the total number of detected cells in a field of view in the first frame of a time-lapse sequence. Then, at least the first 100 cells in a field of view were manually observed and marked at the first instance of movement using the ImageJ cell counter plugin to give total number of gliding cells and time to onset of gliding (or the absence of gliding within the observed 6 h).

### Cloning for protein expression

Nucleotide primer pairs 5′-gtttaactttaagaaggagatatacatatgaatgcag-cagctcagtccgtcg-3′ and 5′-gctgcactaccgcgtggcacaagcttgatcagctttaa-gaagcggcccag-3′ were used to amplify the *B. bacteriovorus* HD100 *bd1971* gene). The product was utilized in a second round of PCR, then inserted into a modified version of the expression plasmid pET41 (Novagen, GST removed, placing a thrombin cleavable His$_8$ tag on the C-terminal end of the protein; left uncleaved) in a restriction-free process. The truncated cNMP domain-only form (AA 1–150) was produced via mutagenesis (151 to stop codon) of a N-terminally tagged variant, which itself was made similarly to the C-tagged form but using pET28a and primer pairs 5′-gcagcggcctggtgccgcgcggcagccatatgaatgcagcagctcagtccgtcg-3′ and 5′-ctcagtggtggtggtggtggtgctcgagttagatcagctttaagaagcggcccag-3′. Constructs were confirmed by sequencing and introduced into the *E. coli* expression strain BL21λDE3.

### Protein production and purification

Cells were grown at 37°C until reaching an OD600 of ~0.6, then gene expression induced with 0.2 mM IPTG for 20 h at 20°C. Harvested cells (approx. 11 g from 2 l 2× LB culture) were re-suspended by tumbling in 45 ml buffer A (20 mM HEPES pH 7.2, 0.3M NaCl, 20 mM imidazole, 5% w/v glycerol, and 0.1% w/v Tween-20) and lysed using sonication. Unbroken cells were pelleted by centrifugation at 6,000 *g* for 20 min, and the lysate clarified by a second centrifugation at 200,000 *g* for 1 h, with the supernatant applied to a 1 ml Hi-Trap His column, pre-equilibrated in buffer A. Fractions were eluted in a stepwise manner, using buffer A containing 40 and 300 mM imidazole. Pure fractions of Bd1971 were dialyzed overnight into buffer B (25 mM Tris pH 7.0, 300 mM NaCl, 8% w/v glycerol, 1 mM EDTA) and concentrated to a protein concentration of ~45 mg/ml (supplemented as required for the varying crystal forms with either 2 mM cAMP or cGMP and either 2 mM $MgCl_2$ or 2 mM $CaCl_2$).

Apo-Bd1971 was prepared for use in isothermal calorimetry experiments, *p*-nitrophenyl phosphatase activity assays, and

crystallization trials. Purified Bd1971 protein was applied to a 1 ml bed volume of adenosine 3′,5′-cyclic monophosphate–agarose (Sigma: A0144) pre-equilibrated with buffer B. This affinity resin was then washed with 50 ml of buffer B prior to elution of protein with buffer C (100 mM CAPS pH 10.5, 300 mM NaCl, 5% glycerol). Fractions containing apo-Bd1971 were dialyzed into buffer D (25 mM Tris pH 8.0, 250 mM NaCl, 2% glycerol, 5% ethanol) and concentrated to 20 mg/ml. The removal of endogenous cyclic nucleotides from Bd1971 was confirmed by the lack of *p*-nitrophenyl phosphatase activity in fractions prepared by this method. Phosphodiesterase activity could be restored by supplementation with 2 mM of either cAMP or cGMP. Apo-Bd1971$^{\Delta EAL}$ was prepared by the same method but was extensively dialyzed into buffer E (20 mM Tris pH 8.0, 150 mM NaCl) and concentrated to 40 mg/ml.

## Structure determination

Crystals were grown by the hanging drop method at 18°C, using 1 μl of protein solution mixed with an equal volume of reservoir solution. Initial crystallization conditions for the full-length cAMP/cGMP-bound forms (using 2.5 mM nucleotide in protein stocks) were identified in 0.2M Mg formate pH 5.5–5.9, 20% w/v PEG 3350; for Bd1971$^{\Delta EAL-mid}$ (AA 1–150, C2 form, 2.5 mM nucleotide) in 0.2 M K nitrate pH 6.9, 20% w/v PEG 3350; for the Bd1971$^{\Delta EAL}$ apo-form (AA 1–150, P4$_1$2$_1$2 form) in 0.1 M Na HEPES pH 7.0, 18% w/v PEG 12000. The c-di-GMP/cAMP/Ca$^{2+}$ ternary complex was obtained using protein supplemented with 2 mM cAMP and 10 mM CaCl$_2$, with growth conditions of 0.1 M MES pH 6.5, 0.2 M Na Acetate, and 15% w/v PEG 8000; these crystals were soaked in mother liquor containing 2 mM c-di-GMP for 40 min at 18°C. Cryoprotection for all forms was attained by sequential addition of increments of mother liquor supplemented with 20% (v/v) ethylene glycol, followed by subsequent flash cooling in liquid nitrogen. Diffraction data were collected at the Diamond Light Source, Oxford. Data were processed using XDS (Kabsch, 2010), and data file manipulations performed using the CCP4 suite of programs (Winn *et al*, 2011). Molecular replacement attempts utilizing PHASER (McCoy *et al*, 2007) with a truncated version of the EAL catalytic domain of *Thiobacillus denitrificans* (PDB code 3N3T; Tchigvintsev *et al*, 2010) yielded a clear solution, identifying two copies in the asymmetric unit. Density from this solution was good enough to place two copies of the cNMP-binding domain from protein kinase G (PDB code 3OCP; Kim *et al*, 2011). The remaining parts of the molecule were built manually using COOT (Emsley & Cowtan, 2004) and model refinement used PHENIX (Zwart *et al*, 2008) and the PDB-REDO server (Joosten *et al*, 2011). The final model is of excellent stereochemical quality, statistics in Table 1.

## Isothermal calorimetry

Apo-Bd1971$^{\Delta EAL}$ was used in isothermal calorimetry experiments to assess the interactions between protein and cyclic nucleotides. The concentrations of Bd1971$^{\Delta EAL}$, cAMP, and cGMP used in these experiments were determined spectrophotometrically using the molar extinction coefficients 2,980 at 280 mm, 14,650 cm$^{-1}$ at 259 nm, and 12,950 at 254 nm, respectively. The binding was

assessed using a VP-ITC microcalorimeter (MicroCal LLC). Bd1971$^{\Delta EAL}$ (1.4 ml) was equilibrated in the sample cell at 25°C, and either cAMP or cGMP prepared in buffer E was added to the syringe. Nucleotide was injected into protein and the heat changes recorded. Following an initial injection of 2 μl, further injections of 6 μl were made with 300 s spacings, this time was sufficient for the signal to return to baseline. Control experiments using the same injection regimen were performed measuring the heat changes associated with the injection of nucleotides into buffer E and also buffer E into protein. Heat changes associated with these control experiments were negligible. The concentration of Bd1971$^{\Delta EAL}$ included in these experiments varied from 50 to 150 μM with cAMP or cGMP at concentrations from 0.5 to 1.6 mM. All experiments were repeated at least twice. Data were interpreted using MicroCal Origin for ITC software (MicroCal LLC).

## Phosphodiesterase activity assays

The phosphodiesterase activity of Bd1971 was assayed using *p*-nitrophenyl phosphate. This general substrate can be hydrolyzed to yield *p*-nitrophenol, which can be quantified spectrophotometrically using the extinction coefficient 18,000 at 405 nm. A stock solution of Apo-Bd1971 protein in buffer D + 10 mM MgCl$_2$ and used to set up a series of reaction tubes supplemented with varying concentrations of cAMP or cGMP. Tubes were equilibrated at 25°C for 30 min prior to the addition of *p*-nitrophenyl phosphate. The final concentrations of components in these reaction mixtures were 1 mg/ml Bd1971, 5 mM *p*-nitrophenyl phosphate and between 0.2 and 1,000 μM cAMP or cGMP. Reaction mixtures were incubated at 25°C. Assays were quenched by the addition of 10 volumes of stop solution (110 mM NaOH, 2.7 mM EDTA, 55 mM K$_2$HPO$_4$), and the concentrations of *p*-nitrophenol were determined in a spectrophotometer using a protein-only reaction mixture as a blank. Assays were also performed using apo-Bd1971, Bd1971$^{R67A}$, or Bd1971$^{\Delta cNMP}$ supplemented with either 10 mM CaCl$_2$, 10 mM MgCl$_2$, or 10 mM EDTA in the presence of 2 mM cAMP or 2 mM cGMP. All assays were performed in triplicate and are representative of two independent experiments.

## HPLC endpoint assays

Bd1971 activity toward c-di-GMP was assessed by HPLC. Reaction mixtures were prepared in buffer D lacking ethanol but supplemented with 10 mM MgCl$_2$ in a final volume of 50 μl. These mixtures contained 100 nM nucleotide-free Bd1971 supplemented with 100 μM cGMP, cAMP, or H$_2$O and were incubated for 30 min at 25°C prior to the addition of 100 μM c-di-GMP. Reactions were terminated by boiling after 15 seconds and then centrifuged to remove any aggregated material. Control reactions with Bd1971 boiled prior to nucleotide addition were also performed, and individual standards of nucleotides, including 100 μM pGpG, were prepared as standards.

Reaction products (5 μl) were separated by HPLC on a Dionex UPLC system with a Kinetex C18 column (Phenomenex; 1.7 μm, 150 × 2.1 mm). Separation was achieved at 45°C and 0.2 ml/min with a convex gradient from 100 mM sodium phosphate buffer pH 5.8–100 mM sodium phosphate buffer pH 5.8, 20% methanol over

10 min. Nucleotides were detected with absorbance at 278 nm, and peaks were identified by comparing to the known standards.

## Statistics

A D'Agostino-Pearson omnibus or Shapiro–Wilks (for samples with fewer datapoints) normality tests were first used to evaluate whether the data were normally distributed. Where data were determined to have a normal distribution, then Student's $t$-test was used, where normality tests failed, the non-parametric Mann–Whitney $U$-test was applied to assess the significance between the data sets. These tests were carried out using GraphPad Prism.

## Data availability

RCSB accession codes for the co-ordinates are as follows: full-length cGMP complex, 6HQ7; full-length cAMP complex, 6HQ4; full-length cAMP and c-di-GMP complex, 6HQ5; apo sensor domain, 6HQ2; half-occupied cAMP-bound sensor domain, 6HQ3.

**Expanded View** for this article is available online.

## Acknowledgements

We would like to thank Elizabeth Arthan and Dr Luke Alderwick for their assistance during the initiation of this work, Nicholas Cotton for advice regarding isothermal calorimetry experiments, Tim Knowles for help with preliminary SAXs analysis, and are grateful to Dr Klaus Futterer for fruitful discussions. We thank Diamond and the Block Allocation Group system for access to synchrotron beamtime. At UoB AL was supported by Leverhulme Trust grant RPG-2014-241 and RM by a BBSRC studentship. PM was supported by a BBSRC Fellowship (BB/S010122/1). At UoN RF and RA were self-funded Masters students, SMB was supported by an NERC PhD studentship NE/I528469/1 and RN by a BBSRC DTP UoN studentship. CL and RT were supported by BBSRC grant BB/M010325/1, LH by Human Frontier Science Programme Grant RGP57/2005 RL by Leverhulme Trust grant RPG-2014-241.

## Author contributions

Experiments were designed by ALL, RES, ITC and LH, and performed with their supervision by all other authors. Protein biochemistry, activity and structure determination were carried out by ITC, RM, MT and WSH in the ALL laboratory; PJM carried out HPLC experiments. Microbiology, gene mutation, gene transcription, B2H analyses and fluorescent and time-lapse microscopy was carried out by SMB, RN, RL, CL, RT, RA, RF and LH in the RES laboratory. The manuscript was drafted by ALL, ITC, CL and RES with contributions and edits from all authors.

## Conflict of interest

The authors declare that they have no conflict of interest.

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
