## [Review Process File · The EMBO Journal]

Nucleotide Signaling Pathway Convergence in a cAMP-sensing bacterial c-di-GMP Phosphodiesterase

Ian T. Cadby, Sarah M. Basford, Ruth Nottingham, Richard Meek, Rebecca Lowry, Carey Lambert, Matthew Tridgett, Rob Till, Rashidah Ahmad, Rowena Fung, Laura Hobley, William S. Hughes, Patrick J. Moynihan, R. Elizabeth Sockett, Andrew L. Lovering

Review timeline:

Submission date:	24 September 2018
Editorial Decision:	12 November 2018
Revision received:	29 March 2019
Editorial Decision:	23 April 2019
Revision received:	11 June 2019
Accepted:	13 June 2019

Editor: Daniel Klimmeck

Transaction Report:

1st Editorial Decision

12 November 2018

Thank you for the submission of your manuscript to The EMBO Journal. Your manuscript has been sent to three referees, and we have received reports from all of them, which I enclose below.

As you will see, the referees acknowledge the potential high interest and novelty of your work, although they also express a number of issues that will have to be addressed before they can support publication of your manuscript in The EMBO Journal. Referee #3 states that the cleavage activity of Bd1971 is not sufficiently demonstrated and requests additional analyses to corroborate this point (ref#3, pt.1). Further, Further, the referees point to issues related to integration of literature, better discussion of the results, missing controls and data illustration would need to be conclusively addressed to achieve the level of robustness needed for The EMBO Journal.

I judge the comments of the referees to be generally reasonable and given their overall interest, we are in principle happy to invite you to revise your manuscript experimentally to address the referees' comments.

Please let me know any time if you have additional questions or need further input on the referee comments.

Please see below for additional instructions for preparing your revised manuscript.

Thank you for the opportunity to consider your work for publication. I look forward to your revision.

REFEREE REPORTS

Referee #1:

The manuscript by Cadby et al. describes the structural and biochemical study of the cyclic-di-GMP phosphodiesterase Bd1971 and the modulation of its activity by cAMP and cGMP. The authors show that it is responsible for reducing [c-di-GMP] but it is not clear whether it is the only enzyme that does that in this bacterium. Coming from outside of the c-di-GMP field, I thought that it was difficult to grasp the importance and role of these phosphodiesterases from the introduction alone. These difficulties were compounded by the use of many undefined abbreviations. Without any improvements, I feel that the manuscript might not appeal to a broad audience.

Semiquantitative studies were done to demonstrate that Bd1971 has Mg-cAMP/Mg-cGMP-dependent activity. It is not clear how robust this activity is. It would be nice to contrast it with activities obtained from related proteins in the discussion. The communication of the allosteric signal from the binding of the regulatory molecule to the active site was frustrated by technical problems with the apo form protein necessitating the generation of a truncation mutant lacking the catalytic domain. This glimpse of the unbound form of the regulatory dimer is used plausibly to generate a model for a relatively complex transition between the activated and unactivated forms of the enzyme.

Aside from problems discussed below, the manuscript was well-written. The work is comprehensive and well-done as far as I can tell.

Abstract: There are non-standard abbreviations that are used but not defined (e.g. EAL, GGDEF). This problem persists in the main text as well.

p. 16, 1/3 down the page: It is not clear what is meant by "...the Vmax recorded in the presence of cGMP was higher than that of cAMP." Are experiments summarized in Figure 5 done with saturating concentrations of the substrate PNPP?

p. 19, 1/2 down the page: The sentence containing "...on binding cAMP could have effect protein-protein interactions." doesn't make sense.

Figure 5: It is not clear what the PNPP concentrations here are. The last line of the figure legend saying that varying concentrations of cAMP/cGMP is also confusing - it conflicts with the first line stating that saturating concentrations were used. An explicit description of all concentrations would be best.

Figure 9: I am not sure but it seems that the data here are somewhat redundant with the positive results in Figure 5. Perhaps they could be combined?

Table 1: High resolution Rmerge is missing for the 6HQ2 structure.

Page numbers would have aided in reviewing this MS.

Referee #2:

The manuscript by Cadby et al. describe the structural and mechanistic analysis of an EAL-containing c-di-GMP phosphodiesterase from *B. bacteriovorus* (Bd1971), submitted to PDB with the code(s) 6HQ*. This is an interesting paper with detailed experiments, interesting analysis and interpretation and extensive references providing context to the specific field. In particular, the high-resolution structure determination of Bd1971 in both active and inactive states is a valuable contribution for a better understanding of this enzyme class.

The paragraphs largely correspond (and document) the relevant figures -- which follow a logical order that reflects the reasoning of the analysis and the sequence of experiments. This is a standard, solid, structural analysis paper that can be published in EMBO J.

PS the documentation of materials and methods used is exemplary. No remarks there.

Some comments, follow, hopefully helpful for the authors to improve their manuscript.

Major comments:

1. Some further comments (perhaps a short paragraph) at the end of the results, integrating a little more the comparative analysis across other species (e.g. the 3N3T structure used in the paper) and the multi-domain nature of this enzyme class might provide the big picture. There is a lot of information already in the manuscript, but the details are lost amongst the structural work. A minimal (cartoon?) figure with the salient features of the multi-domain organisation might also be useful -- leave it up to the authors to consider this suggestion. This might include homologs from *Pseudomonas*, *Mycobacterium* and other organisms.
2. As the interaction analysis is a significant contribution beyond the structural work, a cartoon (perhaps combined with the suggestion above) could show a 'comparative' view, as a discussion point across, say, two species -- what are the orthologs in *Pseudomonas* based on the *Bdellovibrio* proteins (Bd3125, Bd0742) etc. This would be a welcome element.

Minor comments:

1. reviewed herein1 -> reviewed elsewhere? / ref is given at the end of the second sentence, it could be deleted here, potentially.
2. "were described by Galperin and coworkers as having" -> "were described as having".. ? Reads a bit better, as reference is provided.
3. "an N-terminal cNMP domain" -> an N-terminal cNMP binding? / sensory ? domain.
4. "genome sequencing revealed several secondary mutations" -> Nice ,but is this shown?
5. "we have estimated the amino acid register in this region from physical attributes and conservation patterns of the segment, but it remains ambiguous" -- can be explained a bit further.
6. 3N3T from *T. denitrificans*, how is related to 2R6O, detected as the best homolog of Bd1971 (3N3T~2R6O = 100% identity, apparently).
7. the (full-length) -> lose the parentheses. It's full-length.
8. "Our structures" -> the resolved structures...
9. Species names in the reference list could be in italics.

Referee #3:

c-di-GMP is a widespread second messenger signaling molecule in bacteria. C-di-GMP levels are believed to be tightly controlled in order for discretely regulated pathways to coexist in the same bacterial cell, yet the molecular details of how c-di-GMP degrading enzymes are regulated remain poorly understood. Here, Cadby et al. identify Bd1971 as a c-di-GMP degrading enzyme in *Bdellovibrio*, and discover allosteric regulation by a cAMP/cGMP-sensor domain. In an elegant series of structural experiments, the authors determine crystal structures of the full-length Bd1971 in complex with cAMP and c-di-GMP, and a truncated variant of the cNMP-sensor domain (Bd1971- Δ EAL) in apo and half-site occupied conformations. The results allow the authors to develop a model of how the cNMP-sensory domain allosterically controls c-di-GMP degradation.

Overall the data are clearly described and will be of high interest to the field. However, the authors rely solely on the use of a general PNPP phosphodiesterase assay and do not definitively demonstrate the ability of Bd1971 to degrade c-di-GMP. Although the correlative evidence is overwhelmingly strong, and the authors determine a structure of Bd1971 bound to c-di-GMP and inactivating calcium ions, it is important to definitively demonstrate c-di-GMP cleavage activity

especially as assaying c-di-GMP cleavage may reveal more insight into the interesting mechanism of half-site regulation. Additionally, I have minor comments to help improve presentation within the text.

Major Points:

1) The authors should experimentally demonstrate that Bd1971 is capable of cleaving c-di-GMP. The assay is relatively simple, especially given assay conditions are already established, and is particularly important as this is the first description of a functional EAL-like c-di-GMP phosphodiesterase in *Bdellovibrio*.

Minor Comments:

1) The description of a Kanamycin cassette Bd1971 mutant with secondary mutations is confusing and detracts from the narrative. The secondary mutations are not listed, and the manuscript focuses exclusively on the Bd1971 clean knock-out strain that does not recapitulate the increased curvature morphology phenotype.

2) Some aspects of the structural data are currently unclear, and should be better supported with additional figures. In particular, no evidence is shown for how ordered the substrates are in the structure, or images of the actual maps used to assign key features.

A. Include a figure depicting the crystal packing of full-length Bd1971 and the evidence used to trace the linker between the EAL and cNMP-sensory domains. It is unclear from the current text how unambiguous the "domain-swap" conformation is in the density.

B. Include a figure showing omit maps for c-di-GMP and cNMP, and consider including the average protein and ligand B-factor information in Table 1 or the methods to help the reader interpret how well ordered the ligands are in each structure. The text states "clear electron density is observed" but no example images of the map are included.

C. Include an annotated protein alignment of Bd1971 and related homologs, especially to help discussion of conservation in key regions like the P-loop helix and linker regions.

3) How was the resolution cut-off for the full length + cGMP dataset (6HQ7) determined? The current high-resolution shell statistics (Rpim 38.9, I/o 2.9) suggest more high-resolution reflections should be included?

4) The discussion is too long, 7 pages. Many points seem unnecessary as they are already stated clearly in the results and not meaningfully expanded upon in the discussion. The points that are of particular interest (clear statement of the model for how cNMP-regulation occurs in Bd1971, how the new data relates to cNMP/c-di-GMP co-regulation in other systems, and then signaling mechanism in related cNMP-sensory proteins) would be much stronger in a shorter discussion.

Text Suggestions:

- The scale bars in Figure 1B are currently confusing as they are the same color as the mCherry bacteria.

1st Revision - authors' response

29 March 2019

Referee #1:

The manuscript by Cadby et al. describes the structural and biochemical study of the cyclic-di-GMP phosphodiesterase Bd1971 and the modulation of its activity by cAMP and cGMP. The authors show that it is responsible for reducing [c-di-GMP] but it is not clear whether it is the only enzyme that does that in this bacterium.

***Bd1971 is indeed the only EAL enzyme (generating linear pGpG, another signalling molecule) to perform this role in *Bdellovibrio*, but there are HD-GYP enzymes (which will degrade c-di-GMP into 2 x GMP) in the genome and we both state this clearly and reference this in the manuscript: "Candidates for lowering of c-di-GMP levels in *Bdellovibrio* are limited: only two of the six noted**

HD-GYP proteins have the consensus motif for substrate binding⁹, and homology searches identify a single EAL domain phosphodiesterase, Bd1971”.

Coming from outside of the c-di-GMP field, I thought that it was difficult to grasp the importance and role of these phosphodiesterases from the introduction alone. These difficulties were compounded by the use of many undefined abbreviations. Without any improvements, I feel that the manuscript might not appeal to a broad audience.

*The enzymes are not known by abbreviations, they are named by the motifs present in their active site (GGDEF for synthases, EAL for hydrolase class 1, HD-GYP for hydrolase class 2). This is standard in the field and we need to keep this nomenclature to relate to all the other publications on these proteins. We feel that the opening paragraph of the introduction explains this for the non-expert without being overlong, but have now added the text “named after active site motifs” to assist the lay reader.

Semiquantitative studies were done to demonstrate that Bd1971 has Mg-cAMP/Mg-cGMP-dependent activity. It is not clear how robust this activity is.

It would be nice to contrast it with activities obtained from related proteins in the discussion.

*We acknowledge that it is often interesting to compare systems but do not feel that comparisons between Bd1971 activity and other EAL domain proteins would be meaningful. Firstly, the activities we initially determined are with respect towards the generic substrate pNPP, not the native substrate. Secondly, in developing our pNPP assays we found that Bd1971 activity varied greatly with the reaction buffer used (for example, no activity was recorded in phosphate buffer). Comparison of activity between samples of the same protein is certainly meaningful, but comparison between enzymes is rendered uninformative as it is subject to too many variables. To answer the query about robustness, our *in vivo* work demonstrates that Bd1971 activity significantly affects intracellular c-di-GMP levels, and our new *in vitro* HPLC assays secondarily confirm this.

The communication of the allosteric signal from the binding of the regulatory molecule to the active site was frustrated by technical problems with the apo form protein necessitating the generation of a truncation mutant lacking the catalytic domain. This glimpse of the unbound form of the regulatory dimer is used plausibly to generate a model for a relatively complex transition between the activated and unactivated forms of the enzyme.

Aside from problems discussed below, the manuscript was well-written. The work is comprehensive and well-done as far as I can tell.

*We thank the reviewer for this broad appreciation of the work.

Abstract: There are non-standard abbreviations that are used but not defined (e.g. EAL, GGDEF). This problem persists in the main text as well.

*See prior comment on standard usage of these terms.

p. 16, 1/3 down the page: It is not clear what is meant by "...the Vmax recorded in the presence of cGMP was higher than that of cAMP." Are experiments summarized in Figure 5 done with saturating concentrations of the substrate PNPP?

*This sentence was included to highlight that cGMP stimulates Bd1971 activity more than cAMP does. We have now changed this to read "...Bd1971 saturated with cGMP had a higher phosphodiesterase activity than when saturated with cAMP" for clarity. The experiments in Figure 5 were done with saturating concentrations of the effector, either cAMP or cGMP.

p. 19, 1/2 down the page: The sentence containing "...on binding cAMP could have effect protein-protein interactions." doesn't make sense.

*This has now been altered to “We reasoned that the two-step conformational change of Bd1971 upon binding cAMP could have an effect on protein- protein interactions”

Figure 5: It is not clear what the PNPP concentrations here are. The last line of the figure legend saying that varying concentrations of cAMP/cGMP is also confusing - it conflicts with the first line stating that saturating concentrations were used. An explicit description of all concentrations would be best.

**We apologise for this confusion. The concentrations have now been added to the figure legend and the last line has been deleted.*

Figure 9: I am not sure but it seems that the data here are somewhat redundant with the positive results in Figure 5. Perhaps they could be combined?

**Figure 9 relates co-operativity to activity (co-operativity apparent from ITC of figure 8) and chronologically doesn't make sense to be merged with figure 5. We also apologise that we noticed an error in the text of legend to figure 8; two replicates were used not three – this does not affect data interpretation or validity and we have amended the legend accordingly.*

Table 1: High resolution Rmerge is missing for the 6HQ2 structure.

**The value is not missing, but exceeds 100, and it is standard practice to omit the value, referring instead to the more useful Rpim and CC ½ values below.*

Page numbers would have aided in reviewing this MS.

**We agree and apologise to the reviewer for this omission.*

Referee #2:

The manuscript by Cadby et al. describe the structural and mechanistic analysis of an EAL-containing c-di-GMP phosphodiesterase from *B. bacteriovorus* (Bd1971), submitted to PDB with the code(s) 6HQ*. This is an interesting paper with detailed experiments, interesting analysis and interpretation and extensive references providing context to the specific field. In particular, the high-resolution structure determination of Bd1971 in both active and inactive states is a valuable contribution for a better understanding of this enzyme class.

**We thank the reviewer for this positive appraisal.*

The paragraphs largely correspond (and document) the relevant figures -- which follow a logical order that reflects the reasoning of the analysis and the sequence of experiments. This is a standard, solid, structural analysis paper that can be published in EMBO J.

PS the documentation of materials and methods used is exemplary. No remarks there.

**Again, we thank the reviewer and feel this was necessary to document some of the complexity of probing the allostery.*

Some comments, follow, hopefully helpful for the authors to improve their manuscript.

Major comments:

1. Some further comments (perhaps a short paragraph) at the end of the results, integrating a little more the comparative analysis across other species (e.g. the 3N3T structure used in the paper) and the multi-domain nature of this enzyme class might provide the big picture. There is a lot of information already in the manuscript, but the details are lost amongst the structural work. A minimal (cartoon?) figure with the salient features of the multi-domain organisation might also be useful -- leave it up to the authors to consider this suggestion.

**We thank the reviewer for this suggestion but feel that the domain organisation is best represented structurally in figure 3A rather than linearly in a schematic.*

This might include homologs from *Pseudomonas*, *Mycobacterium* and other organisms.

*We agree that a comparison is useful and so have added a new figure (Expanded View 4) detailing the agreement with related homologues from a variety of organisms – none of these are in the strains mentioned above which lack an apparent hybrid cNMP-EAL protein.

2. As the interaction analysis is a significant contribution beyond the structural work, a cartoon (perhaps combined with the suggestion above) could show a 'comparative' view, as a discussion point across, say, two species -- what are the orthologs in *Pseudomonas* based on the *Bdellovibrio* proteins (Bd3125, Bd0742) etc. This would be a welcome element.

*The *Bdellovibrio* GGDEFs (Bd3125, Bd0742) are unique/restricted to related predatory bacteria (no proteins with this domain architecture in *Pseudomonas*) and so we cannot draw any parallels here.

Minor comments:

1. reviewed herein1 -> reviewed elsewhere? / ref is given at the end of the second sentence, it could be deleted here, potentially.

*We agree with reviewer 2 and have deleted the reference to this in the first sentence.

2. "were described by Galperin and coworkers as having" -> "were described as having".. ? Reads a bit better, as reference is provided.

*We have made this alteration as suggested.

3. "an N-terminal cNMP domain" -> an N-terminal cNMP binding? / sensory ? domain.

*We have altered this to read "N-terminal cNMP sensory domain".

4. "genome sequencing revealed several secondary mutations" -> Nice ,but is this shown?

*We have now removed these minor details of the original Kan-selected mutant (an edit of section titled "Disruption or deletion of bd1971 results in higher global c-di-GMP levels"; this now only includes details on the "clean" KO strain, and deletes the original two figures from Expanded View section) as we felt this was a little distracting (see response to reviewer 3), and will provide details of these secondary mutations in another publication.

5. "we have estimated the amino acid register in this region from physical attributes and conservation patterns of the segment, but it remains ambiguous" -- can be explained a bit further.

*A detailed explanation of this is provided in response to reviewer 3.

6. 3N3T from *T. denitrificans*, how is related to 2R6O, detected as the best homolog of Bd1971 (3N3T~2R6O = 100% identity, apparently).

*We could have used either (both are listed in the PDB as coming from the same publication, 2R6O is apo-, 3N3T is ligand complex), but chose 3N3T.

7. the (full-length) -> lose the parentheses. It's full-length.

*We have removed the parentheses.

8. "Our structures" -> the resolved structures...

*We have amended this for the two instances it occurred in the text.

9. Species names in the reference list could be in italics.

*These have all been changed in accordance to this request.

Referee #3:

c-di-GMP is a widespread second messenger signaling molecule in bacteria. C-di-GMP levels are believed to be tightly controlled in order for discretely regulated pathways to coexist in the same bacterial cell, yet the molecular details of how c-di-GMP degrading enzymes are regulated remain poorly understood. Here, Cadby et al. identify Bd1971 as a c-di-GMP degrading enzyme in *Bdellovibrio*, and discover allosteric regulation by a cAMP/cGMP-sensor domain. In an elegant series of structural experiments, the authors determine crystal structures of the full-length Bd1971 in complex with cAMP and c-di-GMP, and a truncated variant of the cNMP-sensor domain (Bd1971- Δ EAL) in apo and half-site occupied conformations. The results allow the authors to develop a model of how the cNMP-sensory domain allosterically controls c-di-GMP degradation.

**We thank the reviewer for “elegant series of structural experiments” and appreciation of our study.*

Overall the data are clearly described and will be of high interest to the field. However, the authors rely solely on the use of a general PNPP phosphodiesterase assay and do not definitively demonstrate the ability of Bd1971 to degrade c-di-GMP. Although the correlative evidence is overwhelmingly strong, and the authors determine a structure of Bd1971 bound to c-di-GMP and inactivating calcium ions, it is important to definitively demonstrate c-di-GMP cleavage activity especially as assaying c-di-GMP cleavage may reveal more insight into the interesting mechanism of half-site regulation. Additionally, I have minor comments to help improve presentation within the text.

Major Points:

1) The authors should experimentally demonstrate that Bd1971 is capable of cleaving c-di-GMP. The assay is relatively simple, especially given assay conditions are already established, and is particularly important as this is the first description of a functional EAL-like c-di-GMP phosphodiesterase in *Bdellovibrio*.

**We agree with the reviewer that this would strongly validate our other assays and so now provide this information via HPLC in a new panel (figure 5B). This new experimentation confirms both direct cleavage of c-di-GMP, and provides extra supporting evidence that cleavage is stimulated by cAMP and cGMP.*

The text added for this confirmation of activity reads “We next tested whether Bd1971 has activity towards the assumed native substrate, c-di-GMP. Reaction mixtures containing c-di-GMP and either apo- or cNMP-supplemented Bd1971 were resolved by HPLC (Figure 5b). Consistent with our predictions, Bd1971 converted c-di-GMP to pGpG. Bd1971 activity with respect to c-di-GMP was increased ~8-fold in the presence of cAMP or cGMP, further supporting the proposal that cNMPs are activators of Bd1971.”

Minor Comments:

1) The description of a Kanamycin cassette Bd1971 mutant with secondary mutations is confusing and detracts from the narrative. The secondary mutations are not listed, and the manuscript focuses exclusively on the Bd1971 clean knock-out strain that does not recapitulate the increased curvature morphology phenotype.

**We agree with the reviewer and so have removed this from the manuscript, altering the text and (original) supplementary figures 1 and 2. We retain the detail of the clean knock-out strain.*

2) Some aspects of the structural data are currently unclear, and should be better supported with additional figures. In particular, no evidence is shown for how ordered the substrates are in the structure, or images of the actual maps used to assign key features.

A. Include a figure depicting the crystal packing of full-length Bd1971 and the evidence used to trace the linker between the EAL and cNMP-sensory domains. It is unclear from the current text how unambiguous the "domain-swap" conformation is in the density.

*We agree that it would benefit confidence in ligand structures to include map features – these have been added to a new figure (Expanded View 1) detailing fo-fc omit density for each complex. The linker density is also included as a panel on this figure (as evidence of tracing). We have also clarified the assignment of a domain swap, adding “the distance between residues 111 and 117 is a more favourable 11Å in a domain swap and a less favourable 21Å for non-swapped assignment” to the text.

B. Include a figure showing omit maps for c-di-GMP and cNMP, and consider including the average protein and ligand B-factor information in Table 1 or the methods to help the reader interpret how well ordered the ligands are in each structure. The text states "clear electron density is observed" but no example images of the map are included.

*See above comment, this information has now been added to a new figure. The density for all nucleotide ligands is strikingly apparent (maps at 3s) and should convince both the reviewer and readership, especially so given similarity of binding pose to that observed in other systems.

C. Include an annotated protein alignment of Bd1971 and related homologs, especially to help discussion of conservation in key regions like the P-loop helix and linker regions.

*We agree and have included this in a new figure (see response to Reviewer 2).

3) How was the resolution cut-off for the full length + cGMP dataset (6HQ7) determined? The current high-resolution shell statistics (Rpim 38.9, I/o 2.9) suggest more high-resolution reflections should be included?

*The resolution / reflection intensity falls off quicker than the outer shell of 2.9 suggests; including “extra” data did not assist or benefit refinement.

4) The discussion is too long, 7 pages. Many points seem unnecessary as they are already stated clearly in the results and not meaningful expanded upon in the discussion. The points that are of particular interest (clear statement of the model for how cNMP-regulation occurs in Bd1971, how the new data relates to cNMP/c-di-GMP co-regulation in other systems, and then signaling mechanism in related cNMP-sensory proteins) would be much stronger in a shorter discussion.

*We respectfully disagree and would like to retain this material, which helps to convey the complexity and relevance of our results; we note that reviewers 1 (“manuscript was well-written.... work is comprehensive”) and 2 (“interesting analysis and interpretation and extensive references providing context to the specific field”) were very positive about our interpretation herein.

Text Suggestions:

- The scale bars in Figure 1B are currently confusing as they are the same color as the mCherry bacteria.

*The scale bar colour has now been altered (white).

2nd Editorial Decision

23 April 2019

Thank you for submitting your revised manuscript for consideration by The EMBO Journal. Your revised study was sent back to two of the original referees for re-evaluation, and we have received comments from both of them, which I enclose below. As you will see the referees find that their concerns have been sufficiently addressed and they are now broadly in favour of publication.

Thus, we are pleased to inform you that your manuscript has been accepted in principle for publication in The EMBO Journal, pending some minor issues related to formatting and data representation, which need to be adjusted at re-submission.

REFEREE REPORTS

Referee #1:

All comments from the initial review were adequately addressed.

Referee #3:

The revised manuscript is significantly improved and addresses all of my comments. I recommend publication.

2nd Revision - authors' response

11 June 2019

The authors performed all requested editorial changes.

Corresponding Author Name: Andrew Lovering

Manuscript Number: EMBOJ-2018-100772